# The conserved HIV-1 spacer peptide 2 triggers matrix lattice maturation

James C. V. Stacey[1,2,9], Dominik Hrebík[1,9], Elizabeth Nand[3], Snehith Dyavari Shetty[4], Kun Qu[5], Marius Boicu[1], Maria Anders-Össwein[4], Pradeep D. Uchil[3], Robert A. Dick[6,8], Walther Mothes[3], Hans-Georg Kräusslich[4,7], Barbara Müller[4] & John A. G. Briggs[1✉]

The virus particles of human immunodeficiency virus type 1 (HIV-1) are released in an immature, non-infectious form. Proteolytic cleavage of the main structural polyprotein Gag into functional domains induces rearrangement into mature, infectious virions. In immature virus particles, the Gag membrane-binding domain, MA, forms a hexameric protein lattice that undergoes structural transition, following cleavage, into a distinct, mature MA lattice[1]. The mechanism of MA lattice maturation is unknown. Here we show that released spacer peptide 2 (SP2), a conserved peptide of unknown function situated about 300 residues downstream of MA, binds MA to induce structural maturation. By high-resolution in-virus structure determination of MA, we show that MA does not bind lipid into a side pocket as previously thought[1], but instead binds SP2 as an integral part of the protein–protein interfaces that stabilize the mature lattice. Analysis of Gag cleavage site mutants showed that SP2 release is required for MA maturation, and we demonstrate that SP2 is sufficient to induce maturation of purified MA on lipid monolayers in vitro. SP2-triggered MA maturation correlated with faster fusion of virus with target cells. Our results reveal a new, unexpected interaction between two HIV-1 components, provide a high-resolution structure of mature MA, establish the trigger of MA structural maturation and assign function to the SP2 peptide.

HIV-1 morphogenesis proceeds through assembly of Gag at the plasma membrane of the infected cell. Gag–Gag, Gag–membrane, Gag–RNA and Gag–host protein interactions drive bud formation and release of an immature, non-infectious viral particle from the cell surface. Concomitant with budding, Gag undergoes an ordered proteolytic cleavage cascade mediated by the viral protease (PR) into its subdomains (MA (matrix), CA (capsid), SP1, NC (nucleocapsid), SP2 and p6; Extended Data Fig. 1a). This leads to structural changes in the protein domains and a marked rearrangement of viral architecture[2–5]. Maturation converts a non-infectious particle optimized for assembly into a virion specialized and competent for entry and infection[2–4,6]. CA maturation has been studied in detail, providing models for how cleavage events upstream and downstream of CA trigger disassembly of the spherical, hexameric immature CA lattice and reassembly into a structurally and functionally distinct mature CA lattice[4,5,7–10].

The MA domain is responsible for the recruitment of Gag to the host plasma membrane during virus assembly[11–14]. MA forms a trimer that interacts with membranes in a phosphatidylinositol 4,5-bisphosphate (PtdIns(4,5)P₂)-dependent manner through a highly basic region (HBR) and an amino-terminal myristate moiety[15,16]. Myristate is initially sequestered within MA, but is exposed following MA trimerization, and can insert into the inner leaflet of the bilayer[15–17] (Extended

Data Fig. 1b). Alterations in MA lead to defects in envelope protein (Env) incorporation, but the mechanisms of Env incorporation are not fully resolved[18–22]. We recently showed that trimer–trimer interactions between N-terminal residues link MA trimers together into a loose hexameric lattice in the immature virion[1]. Following maturation, these interactions are replaced by a larger interface between MA trimers to form a distinct, regularly packed hexameric lattice in the mature virion[1]. In contrast to the case for CA, high-resolution structural data on the immature and mature MA lattices are not available.

Our previous cryogenic electron tomography (cryo-ET) analysis of intact virions showed that a cleft in MA formed by α-helix 4 and the loop between α-helix 1 and α-helix 2 is empty and exposed in the immature virus but occupied and facing the trimer–trimer interface in the mature MA lattice[1]. The density observed in the cleft in the mature virus was consistent with NMR structures of bound, membrane-extracted PtdIns(4,5)P₂ (ref. 12). The same cleft has also been implicated in the binding of host tRNAs[23] by cytosolic Gag (Extended Data Fig. 1b). MA binding to the PtdIns(4,5)P₂-containing plasma membrane would displace tRNA, and free the cleft for binding to (partially membrane-extracted) PtdIns(4,5)P₂ during the later maturation step[1]. Together with observations that both mechanical properties of the viral membrane[24,25] and virion fusogenicity[6,26] change following maturation, our

[1]Department of Cell and Virus Structure, Max Planck Institute of Biochemistry, Martinsried, Germany. [2]Structural Studies Division, MRC Laboratory of Molecular Biology, Cambridge, UK. [3]Department of Microbial Pathogenesis, Yale University School of Medicine, New Haven, CT, USA. [4]Department of Infectious Diseases, Virology, Heidelberg University, Heidelberg, Germany. [5]Infectious Diseases Translational Research Programme, Department of Biochemistry, Yong Loo Lin School of Medicine, National University of Singapore, Singapore, Singapore. [6]Department of Molecular Biology and Genetics, Cornell University, Ithaca, NY, USA. [7]German Center for Infection Research, Heidelberg, Germany. [8]Present address: Department of Pediatrics, School of Medicine, Emory University, Atlanta, GA, USA. [9]These authors contributed equally: James C. V. Stacey, Dominik Hrebík. ✉e-mail: briggs@biochem.mpg.de

results led to the hypothesis that MA maturation changes viral bilayer properties by extracting lipids.

A straightforward assumption would be that maturation of MA is induced when it is released as a mature protein domain from Gag by cleavage between MA and CA. However, we previously observed that this is not the case: a Gag mutant in which cleavage between MA and CA and cleavage between CA and SP1 are blocked (MA–SP1) exhibits an immature CA lattice and a mature MA lattice, suggesting that long-range interactions may be important for MA lattice maturation.

Here we have obtained high-resolution structures of the MA lattice within virus particles and have studied Gag cleavage mutants. These data show that the previously described mature MA ligand is not a lipid; instead, it is SP2, a highly conserved downstream Gag peptide of unknown function. Proteolytic release and MA binding of SP2 is the trigger for MA maturation and correlates with the virus gaining fast, wild-type (WT)-like fusion kinetics.

## High-resolution in-virus MA structures

To determine high-resolution structures of the immature and mature MA lattices within intact viruses, we applied in situ single-particle analysis. We and others recently used a similar approach to determine the structure of the mature HIV-1 CA lattice in vitro at high resolution[27–29]. Single-particle analysis is faster than our previous cryo-ET approach[1], allowing us to collect and analyse larger datasets. A data processing scheme combining very large datasets, 2D and 3D classification steps and use of lattice geometries to predict the positions of additional particles allowed us to overcome confounding densities from other viral components. We obtained reconstructions of the MA lattice from immature (PR$^-$) virus particles at 5.8 Å, from mature virus particles at 3.8 Å and from MA–SP1 virus particles (in which MA also adopts a mature lattice) at 3.1 Å (Fig. 1c–j and Extended Data Figs. 2–6). The lower resolution of the immature lattice probably reflects its higher flexibility. In all cases, the results were consistent with our previous lower resolution cryo-ET structures at the resolution at which they could be compared[1]. However, the substantially higher resolution of the structures described here unravelled crucial structural details.

In the mature MA lattice, the ordered loop between α-helix 1 and α-helix 2 sits at the centre of the inter-trimer interaction, forming a 230-Å$^2$ interface with, in particular, α-helix 4 in its two-fold-related neighbour and a 90-Å$^2$ interface including the $3_{10}$ helix of another MA molecule. The loop between α-helix 1 and α-helix 2 overlaps with the previously described HBR of MA, and the structure revealed HBR binding into an acidic surface in the adjacent trimer: the positions of residues R20, K26 and K95 allow them to form inter-trimer salt bridges with E52, E73 or E74 and E55, respectively (Fig. 1k,l). Sandwiched between the two-fold-related neighbours is the ligand-bound side pocket where the two symmetry-related ligand densities come into close contact, bridged by residue R22 (Fig. 1l,m).

The shape of the observed density for the ligand bound to the MA side pocket (Fig. 1n) was in poor agreement with the binding conformation of PtdIns(4,5)P$_2$ as previously observed by NMR[12]. We therefore explored alternative conformations of PtdIns(4,5)P$_2$, but were unable to obtain a fit for PtdIns(4,5)P$_2$ or for other stoichiometrically relevant lipid species consistent with the shape of the observed density (Extended Data Fig. 7a).

## Influence of Gag processing on MA

The observation that the HIV-1 variant MA–SP1 exhibits a mature MA lattice despite having an immature CA lattice suggests that it is not the separation of MA from CA but other proteolytic cleavage(s) in the carboxy-terminal region of Gag between SP1 and p6 that are relevant for MA lattice maturation. To identify processing steps involved in MA lattice maturation, we produced virus particles for four additional cleavage site mutants: MA–SP2, MA–NC, MA–SP1:NC–p6 and NC–p6 (Fig. 2a and Extended Data Fig. 2a), and characterized their CA and MA lattices by cryogenic electron microscopy (cryo-EM; Fig. 2b and Extended Data Fig. 2b). As expected[8], MA–SP2, MA–NC and MA–SP1:NC–p6 exhibited immature CA lattices, whereas NC–p6 had a mature CA lattice.

To assess the maturation state of the MA lattice, we analysed 2D class averages of regions of the particle edges that showed clear, repetitive MA densities and measured the repetitive spacing of the MA layer using a Fourier analysis (Fig. 2b,c and Extended Data Fig. 8). This measurement is reference-free and independent of 3D alignment. The approximate hexamer–hexamer distance measured from the immature and mature lattice structures was 10.1 nm and 9.0 nm, respectively. The immature and mature class averages showed a Fourier peak at a spatial frequency of 0.030 Å$^{-1}$ or 0.034 Å$^{-1}$, respectively (Fig. 2c), corresponding to the frequency predicted for the [2,1] lattice reflection of the respective lattices (Extended Data Fig. 8c). MA–SP1, as expected, also showed the 0.034 Å$^{-1}$ peak (Fig. 2c) corresponding to the mature MA lattice. We performed the same analysis on images of the additional cleavage mutants and found that MA–NC has a mature MA lattice spacing, whereas MA–SP1:NC–p6, NC–p6 and MA–SP2 exhibited immature MA lattice spacings (Fig. 2c).

The finding that the cleavage mutants MA–SP1:NC–p6 and NC–p6 had retained an immature MA lattice (Fig. 2c) indicated that cleavage within the NC–SP2–p6 moiety, rather than separation of this moiety from MA–SP1, is required for MA lattice maturation. The observation that MA–NC had a mature MA lattice structure, whereas MA–SP2 had an immature MA lattice, indicated that release of the SP2 peptide correlates with MA maturation.

## SP2 release correlates with fast fusion

Efficient HIV-1 fusion was previously shown to depend on Gag maturation[6,26]. We therefore used the same Gag cleavage mutants to measure the effect of MA maturation on viral fusion. We used a split nanoluciferase complementation assay[30] with one enzyme fragment incorporated into virus-like particles and the other into target cells. Particles were added to the target cells, luminescence (fusion) was measured over time, and the time at which each sample reached 50% of total fusion ($T_{1/2}$) was calculated. We found that membrane fusion of immature particles (produced in the presence of protease inhibitors) was twofold slower than observed for WT mature particles (Fig. 2d and Extended Data Fig. 9). Using the Gag cleavage mutants, we found that membrane fusion remained slow when SP2 was not released, whereas fusion was accelerated twofold to the level of WT particles when SP2 was released from Gag (Fig. 2d). The release of SP2 and MA maturation thus correlate with WT-like fusion kinetics.

## Free SP2 binds to MA within the virion

The analyses of cleavage mutants raised the question of how release of the distant SP2 peptide may affect maturation of MA bound to the plasma membrane. Speculating that direct binding of SP2 to MA could be the trigger of MA maturation, we investigated whether the density previously observed in the side pocket of mature MA[1], initially presumed to correspond to a lipid, could instead represent the SP2 peptide. To assess this we first built continuous stretches of amino acids from SP2 into the density. We found that, indeed, the six sequential C-terminal residues (GRPGNF) of SP2 could be confidently fitted into the cryo-EM density in a manner consistent with reconstructions of the mature MA lattice from WT, MA–SP1 and MA–NC particles (Fig. 3a,b and Extended Data Fig. 7b). Second, we used fully automated model building (RELION-5: ModelAngelo) with full-length Gag as the input sequence. As ModelAngelo does not take symmetry into account, we analysed sequences built into each of the three symmetry-related densities

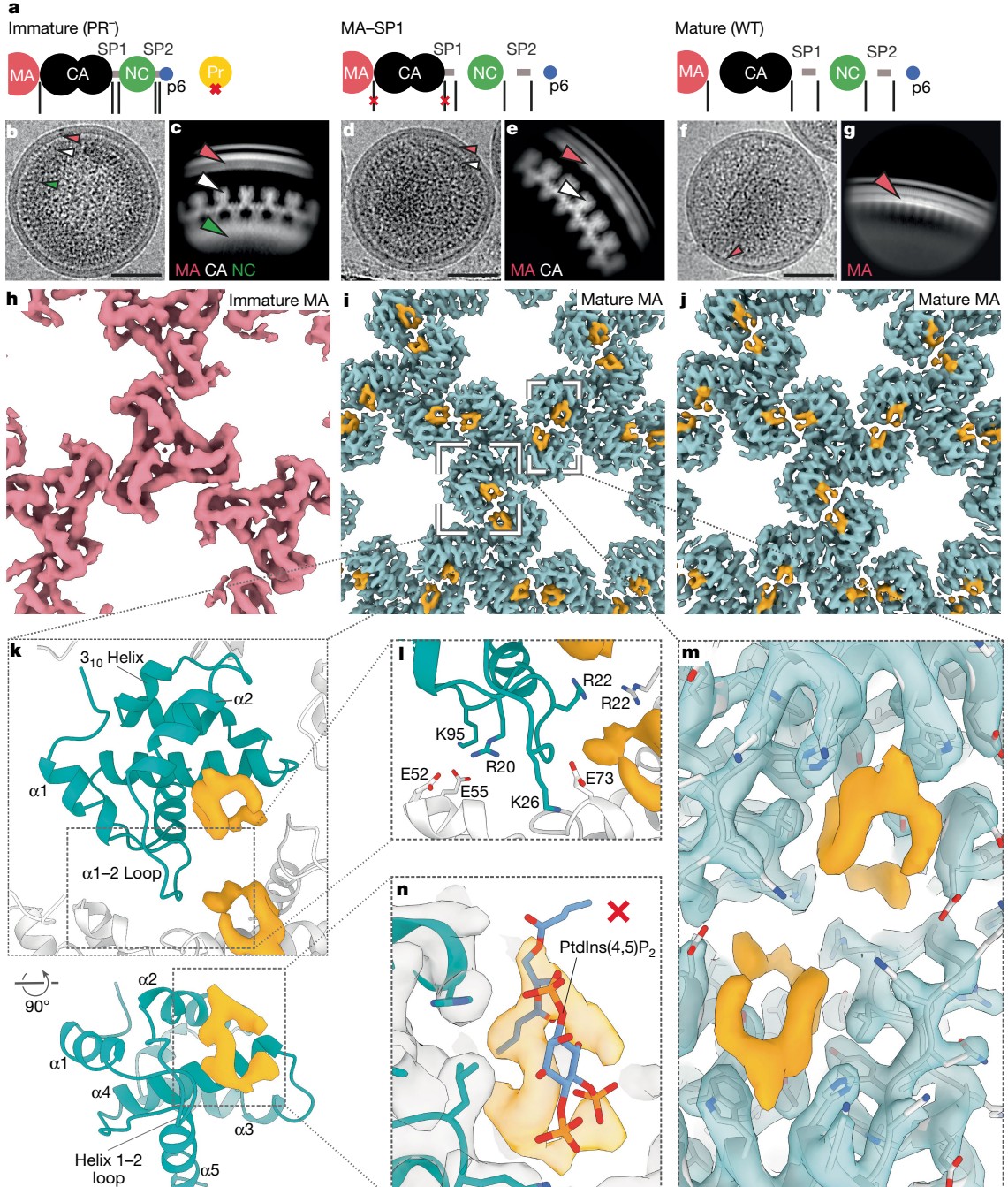

**Fig. 1 | High-resolution reconstructions of MA in immature, mature and MA–SP1 particles. a**, A schematic representation of the Gag cleavage state for each sample. **b**, Representative image of immature (PR⁻) HIV-1 virion from cryo-EM data (Extended Data Table 1). Arrowheads highlight specific Gag domain layers (red, MA; white, immature CA; green, NC). **c**, Side-view 2D class of the Gag layer. Arrowheads as in **b. d**, Representative MA–SP1 HIV-1 virion (Extended Data Table 1). **e**, Side-view 2D class of the Gag layer. As expected, MA and immature CA layers are seen, but NC has been cleaved. Arrowheads as in **b**. **f**, Representative mature (WT) HIV-1 virion (Extended Data Table 1). **g**, Side-view 2D class of the MA layer. Only the MA layer is visible because CA has been cleaved and undergone maturation. **h**–**j**, 3D cryo-EM reconstructions of the MA lattices, viewed from the membrane. As previously observed[1], the MA lattice was found to be immature in immature (PR⁻) particles (**h**), and mature in MA–SP1 (**i**)

and mature (WT) (**j**) particles. Additional ligand density is observed (orange) in both mature MA lattices, but was absent in the immature lattice. **k**, Top: top view of the model of the mature MA lattice, including the ligand density. α1, α-helix 1. One MA monomer is blue; surrounding monomers are white. Bottom: rotated view with helices labelled. **l**, Zoom in of the trimer–trimer interface in the atomic model of the mature MA lattice. The interface is mediated by electrostatic interactions between HBR residues and an electronegative patch along the intra-trimeric interface. **m**, Zoomed-in view of **i** showing ligand binding across the trimer–trimer interface with fitted atomic model (grey). **n**, Rigid fitting of the deposited model of MA bound to PtdIns(4,5)P₂ (Protein Data Bank accession code 2H3Q)[12] shows that the ligand density is not consistent with PtdIns(4,5)P₂ bound as previously reported (the PtdIns(4,5)P₂ atomic model is not accommodated within the orange EM density). Scale bars, 50 nm.

of the central MA trimer independently. In all three instances, the side-pocket density was predicted to correspond to the C-terminal residues of SP2 residues, and the models faithfully recapitulated our initial

modelled conformation (Extended Data Fig. 7c,d). Third, we performed the same automated model building without any input sequence, and again in one position obtained the peptide sequence GRPGNF

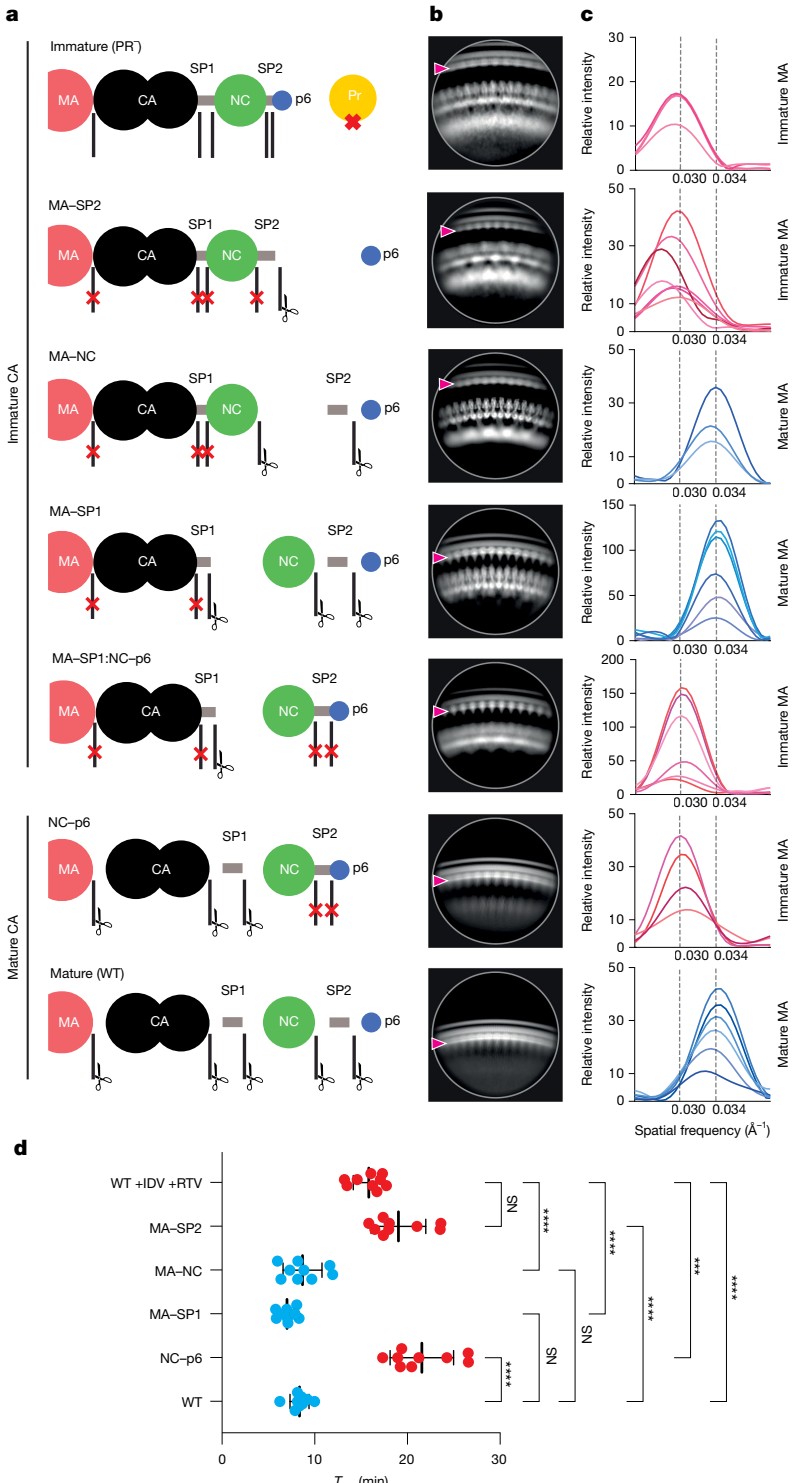

**Fig. 2 | MA lattice states and fusion kinetics for tested Gag cleavage mutants. a**, Schematic representations of the expected Gag cleavage state in all tested HIV-1 constructs. **b**, Representative side-view 2D classes of Gag. Clear ordering is observed within the membrane-bound MA lattices (red arrowhead). The observed MA, CA and NC densities match those expected on the basis of the cleavage state. **c**, Fourier analysis of MA lattice spacings from side views of 2D class averages of HIV cleavage mutants. A Fourier peak with Miller indices of $h,k = [2,1]$ at a spatial frequency of 0.030 Å$^{-1}$ represents the lattice spacing of an immature lattice, whereas a Fourier peak with the same Miller indices at a spatial frequency of 0.034 Å$^{-1}$ represents the lattice spacing of a mature lattice (Extended Data Fig. 8). Each curve represents a single 2D class average. **d**, Histogram of fusion $T_{1/2}$ for the indicated virus-like particles, as in **a**–**c**. All particles contain WT JRFL Env. The top row shows data for immature particles

with WT Gag produced in the presence of protease inhibitors indinavir (IDV) and ritonavir (RTV). Plotted points are from three biological replicates with three technical replicates per biological replicate for $n = 9$. Each biological replicate represents one aliquot from a bulk preparation of viral particles. Statistics were performed using an ordinary one-way analysis of variance (ANOVA) test with Tukey's multiple comparisons tests. Brown–Forsythe test and Bartlett's test were performed as corrections. Bars are mean ± s.d. NS, not significant, $P \geq 0.10$; ***$P < 0.001$; ****$P < 0.0001$. Specific $P$ values: WT +IDV +RTV versus NC–p6, $P = 0.0001$; WT +IDV +RTV versus MA–SP2, $P = 0.1096$; WT versus MA–SP1, $P = 0.9386$; WT versus MA–NC, $P > 0.9999$. Red and blue points represent cleavage mutants observed in **c** as having immature or mature MA, respectively. See also Extended Data Fig. 9.

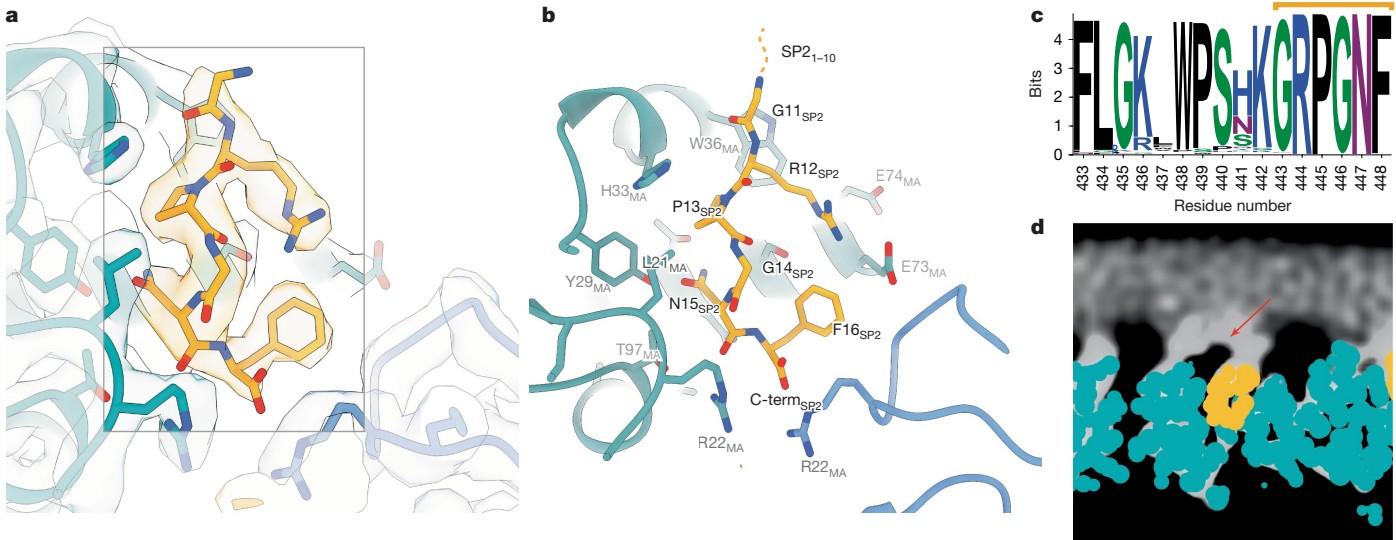

**Fig. 3 | Structure of the SP2 C terminus bound to the mature MA lattice.** **a**, Fit of the six C-terminal SP2 residues (orange) and MA (blue) within the reconstruction of the mature MA lattice of MA–SP1, showing cryo-EM density. **b**, Same view as **a**, with MA and SP2 residues labelled. **c**, WebLogo representation of the conserved amino acid sequence of SP2, generated using the filtered web sequence database at https://www.hiv.lanl.gov. Orange bar indicates the six C-terminal residues that are resolved bound to the MA side pocket. **d**, Additional density (red arrow), continuous with that of bound SP2, is observed above MA and contacting the inner leaflet of the viral membrane. An orthoslice of the unsharpened map, with MA and SP2$_{11-16}$ atomic coordinates overlaid as spheres (blue and orange, respectively), is shown.

in a conformation closely matching to our final model (Extended Data Fig. 7c,d).

In the resulting model, bound SP2 adopted a largely extended conformation, with G11 at the top close to the viral membrane, and the C-terminal F16 at the base (Fig. 3b). The peptide is held in place by multiple interactions, including π-stacking between the backbone amides of SP2 R12 and P13 with MA W36 and H33, respectively, and a salt bridge between SP2 R12 and MA E73. A cleft consisting of residues L21, Y29, S77, T81 and T97 supports the binding of SP2 residues G14 and N15 (Extended Data Fig. 10a–f). Unlike the observed binding mode of tRNA$^{Lys3}$ (ref. 23) and PtdIns(4,5)P$_2$ (ref. 12), R76 and K27 do not form an electrostatic interaction with SP2, adopting a markedly different conformation (Extended Data Fig. 10g–i). SP2 seems to contribute to formation of the inter-trimer interactions in the mature MA lattice through a salt bridge between the C-terminal carbonyl and R22 in the two-fold-related MA molecule (Extended Data Fig. 10c).

The ten N-terminal residues of SP2 were not resolved in our structures. This is not due to proteolytic cleavage within SP2, as the presence of the 16-amino-acid peptide in virions has been demonstrated[31,32]. The orientation of the peptide would position SP2 residues 1–10 at the viral membrane. Re-examination of the mature MA lattice reconstruction revealed additional diffuse density directly above MA monomers in all three reconstructions (WT, MA–SP1 and MA–NC; Fig. 3c). This density is tightly associated with the membrane inner leaflet; we reason that it at least partially represents the N terminus of SP2 (Fig. 3c,d).

## In vitro reconstitution of MA maturation

The findings described above indicate that the SP2 peptide is necessary to induce MA lattice maturation. We next investigated whether SP2 is sufficient to induce formation of a mature MA lattice using an in vitro-reconstituted system. For this, we added purified, recombinant myristoylated MA protein in the presence or the absence of SP2 to lipid monolayers with a composition mimicking the inner leaflet of an HIV-1 membrane[33,34]. Three independently prepared MA-coated lipid monolayers formed on holey carbon EM grids were imaged by cryo-EM for each condition. For each grid, 30 images were collected

from each of 5 randomly selected grid squares. A grid of sub-images was extracted from each image resulting in 139,500 sub-images per grid that were subjected to 2D classification. Classes showed no protein lattice, immature-like MA lattices or mature-like MA lattices (Fig. 4a–c and Extended Data Fig. 11). The total number of sub-images contributing to immature and mature-like MA lattice classes was quantified for three independent experiments (Fig. 4d). In the absence of SP2, 7% of sub-images contributed to immature-like classes and 7% contributed to mature-like classes. The remaining sub-images were assigned to classes containing no interpretable MA lattice. By contrast, 47% of sub-images in the presence of SP2 contributed to mature-like MA classes, whereas no immature-like classes were observed.

We generated high-resolution 2D class averages for the mature-like MA lattices formed in the presence or absence of SP2 (Fig. 4e,f), and compared these to simulated projections of MA structures containing or lacking SP2 (Fig. 4g,h). Mature-like MA lattices formed in the presence of SP2 contain an additional density in a position exactly corresponding to that of SP2 in the cryo-EM structure. Together, these results demonstrate that SP2 can induce formation of mature, membrane-bound MA lattices by binding to the described cleft in the absence of any other components.

## Discussion

The role of the SP2 peptide and its two flanking cleavage sites in the virus life cycle has been an enduring question. Our data now indicate that after its proteolytic release from Gag, SP2 serves as the trigger for MA lattice maturation and becomes an integral component of the inter-trimeric contacts that form the mature MA lattice. In our previous model, MA maturation was coupled to binding of the same pocket by the lipid PtdIns(4,5)P$_2$, which would thereby become partially extracted from the lipid bilayer, providing an explanation for reported changes in the mechanical properties of the virus following maturation[24,25]. The data presented here place the N-terminal residues of SP2 at the viral envelope in a position likely to directly interact with the inner leaflet. The N-terminal residues include lysines, which could interact with lipid headgroups, and hydrophobic residues, which could insert into the bilayer. This positioning would allow SP2 to modulate viral membrane

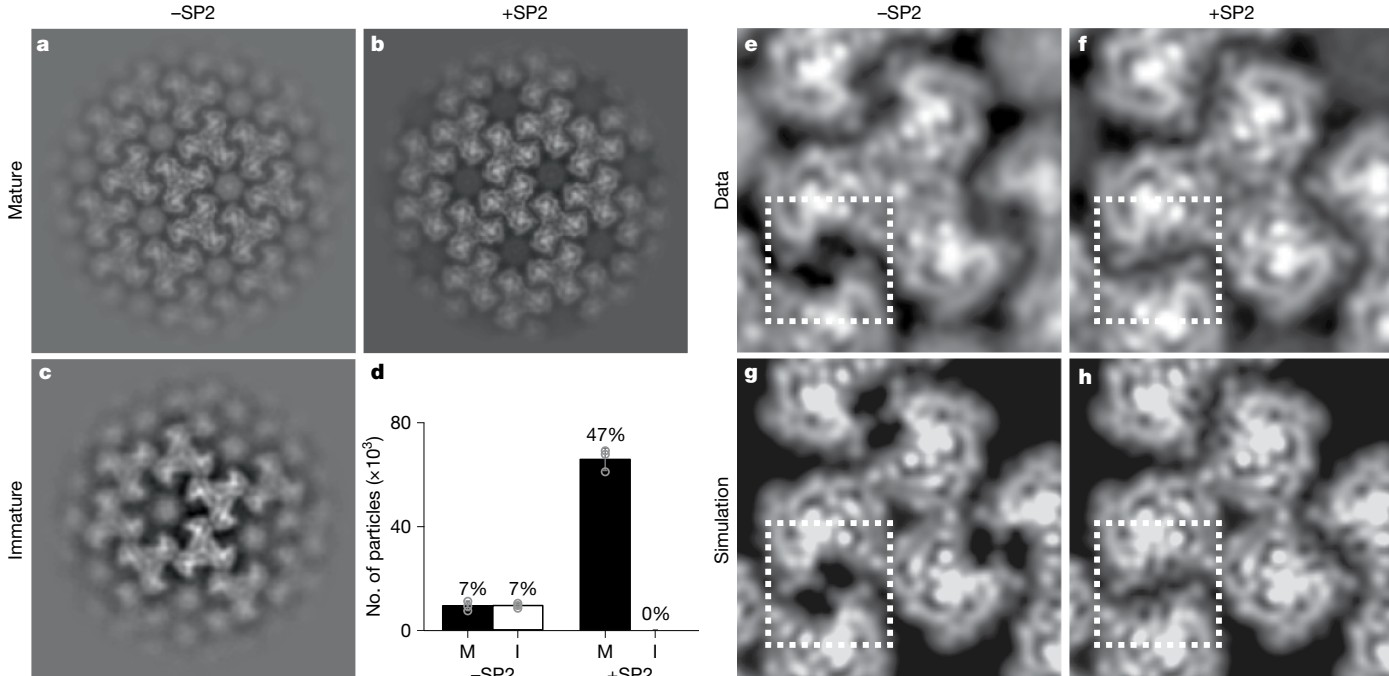

**Fig. 4 | 2D crystallography of HIV-1 MA shows SP2 binding in vitro.**
**a**–**c**, Example 2D class averages of MA assembled on lipid monolayers with or without addition of SP2 showing mature (**a**,**b**) and immature (**c**) arrangements. **d**, Quantification of particles that were identified as either mature (M) or immature (I) MA lattices by 2D classification. A total of 139,500 particles were picked initially for each experiment. Error bars indicate mean ± s.d. between three experiments. **e**,**f**, An extra density in the side binding pocket of MA highlighted by the dashed square is visible in the +SP2 2D class average (**f**). **g**,**h**, Simulated projections of cryo-EM density map of the MA from MASP1 HIV-1 cleavage mutant. The side pocket density corresponding to SP2 was removed in the structure used to generate **g** to simulate the absence of the SP2 peptide.

properties and thereby explain reported changes in virus mechanical properties following MA maturation.

The highly conserved SP2 sequence (Fig. 3c), the two strictly conserved SP2-flanking PR cleavage sites and the structure and stability of the mature MA lattice indicate that SP2 cleavage and MA maturation carry a replicative advantage for the virus. The sequence conservation of the SP2-encoding region can be partly accounted for by coding requirements—this part of the HIV-1 genome includes two overlapping reading frames (*gag* and *pol*) and must maintain an RNA hairpin that promotes ribosomal frameshifting[35]. There is also some evidence that, before proteolytic cleavage, interactions between SP2 and NC may contribute to fine-tuning of NC–nucleic-acid binding[36–38]. We find that release of SP2 and maturation of MA correlate with the virus gaining full fusion competence. However, the HIV-1 mutant NC–SP2, which is defective in release of SP2, does not exhibit an obvious infectivity defect in standard cell culture assays[32,36,39]. We reason that slower fusion kinetics when MA is in an immature state may be limiting in primary cells with lower receptor and co-receptor levels, thus providing an evolutionary advantage to fast fusion. The generation of a highly structured and conserved mature MA layer suggests that it could have additional roles in the post-maturation stage of the HIV-1 replication cycle, potentially regulating early post-entry events. Fully understanding the functional relevance of MA maturation for HIV-1 spread and pathogenesis will require studies in complex systems more closely related to in vivo infection conditions.

The pocket where SP2 binds to MA is versatile, able to bind tRNA[23] as well as PtdIns(4,5)P$_2$ and SP2 (refs. 12,40). Considered together with our observations, this suggests that exchange of ligands in this pocket determines changing functions of MA during the viral replication cycle. In the cytosol of the virus-producing cell, the pocket is bound by tRNAs. tRNAs are displaced from the pocket upon binding of MA to the lipid bilayer during virus assembly. Subsequent binding of cleaved SP2 into the pocket in the released virion then induces MA rearrangement.

SP2-mediated MA maturation shows intriguing parallels with regulation of CA lattice maturation by the other Gag spacer peptide, SP1. SP1 is a structural component of the immature Gag lattice, and release of SP1 by proteolytic maturation promotes transition to the mature CA lattice[8,9,41]. HIV-1 thus has evolved two spacer peptides in its structural polyprotein Gag, which, through proteolytic cleavage at the final steps of Gag processing, serve central functions in the conversion of the immature virus particle into the mature, infection-competent virion. SP1 stabilizes the immature Gag lattice and its proteolytic cleavage promotes formation of the mature, cone-shaped capsid; proteolytic cleavage of SP2 promotes formation and stabilization of the mature MA lattice (Extended Data Fig. 12).

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

# Methods

## Plasmids

All plasmids were based on the subviral plasmid pcHIV[42] that encodes all proteins of HIV-1$_{NL4-3}$ except for Nef. The pcHIV variants MA−SP1, NC−p6, MA−SP1:NC−p6, MA−NC and PR⁻ have been described before[8,39,43]. MA−SP2 was created by introducing alterations at the NC−SP2 cleavage site[32] into pcHIV(MA−NC) by overlap PCR using oligonucleotides 5′-GAGAGACAGGCTTCTTTTTTAGGGAAGACCT GGCCTTCCCACAAGGG-3′ and 5′-CCCTTGTGGGAAGGCCAGGTCTT CCCTAAAAAAGAAGCCTGTCTCTC-3′.

## Cell lines and virus particle production

HEK293T cells (Research Resource Identifier CVCL_0063) were grown in Dulbecco's modified Eagle's medium, 100 U ml⁻¹ penicillin, 100 µg ml⁻¹ streptomycin and 10% fetal calf serum. Genetic characteristics were confirmed by PCR-single-locus-technology, and cells were regularly tested negative for mycoplasma contamination. At 80% confluency, cells were split 1:3 into T175 flasks (CELLSTAR, Greiner BIO-ONE) the day before transfection. Cells were transfected with pcHIV (70 µg per T175 flask) in three T175 flasks per variant using a standard calcium phosphate transfection procedure. At 48 h post transfection, tissue culture supernatant was collected and cleared through a 0.45-µm-pore filter. The filtered supernatant was layered on top of a 20% (w/v) sucrose cushion and subjected to ultracentrifugation at 107,000$g$ for 1.5 h at 4 C. The pellet was resuspended in PBS and stored in aliquots at −80 °C. For quantification, particle-associated reverse transcriptase activity was determined using the Sybr Green Product Enhanced Reverse Transcription assay[44].

## Immunoblotting

Particles were separated by SDS−polyacrylamide gel electrophoresis (20%; acrylamide: bisacrylamide 30:1). Proteins were transferred to a nitrocellulose membrane (Millipore) by semidry blotting and stained with the indicated antisera in PBS with Intercept (PBS) Blocking Buffer (LICORBIO) (sheep anti-CA, polyclonal, 1:5,000 (in-house); rabbit anti-NC, polyclonal, 1:400 (in-house)), followed by corresponding IRDye secondary antibodies in PBS with Intercept (PBS) Blocking Buffer (LICORBIO) (donkey anti-sheep, 1:10,000 (LiCOR Biosciences); and donkey anti-rabbit, 1:10,000 (Rockland)). Detection was performed using a LiCOR Odyssey CLx infrared scanner (LiCOR Biosciences) according to the manufacturer's instructions. Blots are shown in Extended Data Fig. 2a.

## Cryo-EM

All cryo-EM samples of purified HIV-1 particles were prepared and imaged similarly. Purified virus was diluted in PBS buffer in 1:3 (v/v) ratio. A 3 ml volume of the diluted virus sample was applied on a glow-discharged Quantifoil 2/2 holey carbon grid, Cu 300 mesh (Quantifoil Micro Tools), and plunge-frozen into an ethane/propane 1:1 mixture using the Leica EM GP2 (100% humidity, blot time 3.5 s, 20 °C). Grids were loaded into a Titan Krios G4 transmission electron microscope operated at 300 kV, equipped with a CFEG electron source, a Falcon4i direct detector camera and a Selectris X energy filter (ThermoFisher Scientific). Images were collected in electron-event representation (EER) format[45] using EPU (v3) (ThermoFisher Scientific). All datasets were collected at a magnification of ×130,000, resulting in a pixel size of 0.95 Å, with acquisition times ranging from 3.75 to 4.15 s and with a total dose of 40 e⁻ Å⁻². Detailed data acquisition parameters and the number of micrographs for all datasets are given in Extended Data Table 1.

EER videos were rendered as an 8,000 × 8,000 grid and further Fourier-cropped into a 4,000 × 4,000 grid using RELION-4.0 (ref. 46). The videos were motion-corrected, dose-weighted and averaged using RELION-4.0 MotionCorr2 algorithm[47]. Frames were dose-fractionated into groups resulting in a dose of 0.8 e⁻ Å⁻² per fraction. Contrast transfer function (CTF) estimation was performed using the patchCTF algorithm in cryoSPARC v3.3 or v4.4 (ref. 48). Particle picking was performed using crYOLO (v1.7.6)[49]. For each dataset, a new model was trained in crYOLO using a training dataset annotated in a randomly selected set of 50−100 micrographs. Picking models were trained to distinguish virus from background by manually and indiscriminately covering the complete surface of the virus with picks using the crYOLO boxmanager GUI (Extended Data Figs. 3−6).

## Cryo-EM data processing for PR⁻ mutant

The data processing pipeline for PR⁻ MA is summarized in Extended Data Fig. 3. A total of 9,942 motion-corrected micrographs and 3,568,755 particle positions were imported into cryoSPARC v4.4 (ref. 48). Particles, which were picked from large numbers of arbitrary positions over the virus surface, were extracted with a box size of 480 × 480 pixels and Fourier-cropped to 180 × 180 pixels. The first step was to perform a high-resolution reconstruction of the well-ordered immature CA layer to generate CA positions and orientations for use as priors to initialize the refinement of the MA layer. Two rounds of 2D classification were performed in which classes showing side and top views of the immature CA layer were selected (1,630,811 particles) and subjected to heterogeneous refinement with 8 classes, $C_6$ symmetry imposed, and using a previously solved immature structure of an in vitro-assembled HIV capsid (Electron Microscopy Data Bank (EMDB) accession code EMD-3782) as a starting reference 50. Classes showing resolved secondary structures in the CA layer with visible densities representing MA and membrane layers were selected (805,005 particles), re-extracted with a box size of 480 × 480 pixels, and Fourier-cropped to 416 × 416 pixels. Duplicate particles were removed on the basis of a spatial separation, and the accepted particles (676,036 particles) were subjected to three further rounds of 3D refinement with local spatial and angular searches (local refinement), $C_6$ symmetry imposed, and a mask comprising the immature CA layer. Local and global CTF refinements combined with Ewald sphere correction as implemented in cryoSPARC v4.4 were performed in between the 3D refinements. Afterwards, the particles were imported to RELION-4.0, re-extracted with a box size of 480 × 480 and Fourier-cropped to 416 × 416 pixels. The particles were then subjected to Bayesian polishing in RELION-4.0 (ref. 51). The polished particles were imported back to cryoSPARC v4.4 and subjected to a final local refinement resulting in a final high-resolution focused map of the immature CA layer.

The coordinates of the CA layer were then used to predict initial coordinates for the MA layer. To do this, the particles from the CA reconstruction (which is centred on the six-fold symmetry axes) were symmetry-expanded using six-fold symmetry and the 3D coordinates of the centre of the box were shifted to define the positions of the six surrounding three-fold axes of the MA layer. The shifted particles (3,920,498 particles) were then extracted using the new box centre with a box size of 512 × 512 pixels and Fourier-cropped to 256 × 256 pixels. Duplicate particles were removed, and the accepted particles were reconstructed with $C_3$ symmetry. The resulting map was then used to subtract densities corresponding to the immature CA layer from the particles. Subtraction was necessary for the immature MA because the immature CA layer otherwise dominated the reconstruction. The subtracted particles were subjected to three rounds of heterogeneous refinement in cryoSPARC v4.4, with $C_3$ symmetry imposed and eight classes using a cryo-ET-derived reconstruction of the immature MA lattice (EMD accession code EMD-13087) low-pass-filtered to 10 Å as a starting reference 1. Particles belonging to classes that showed aligned MA and membrane layers were selected in each iteration. Selected particles (120,104 particles) were then subjected to non-uniform refinement[52] with $C_3$ symmetry and a mask comprising both the MA and membrane layers followed by a local 3D refinement with a mask comprising only the MA layer.

Next, the dataset was expanded by using the refined positions of MA trimers to predict the positions of neighbouring MA trimers (lattice expansion). To do this, the particles were symmetry-expanded with $C_3$ symmetry (generating the two additional symmetry-related copies of each particle), the centre of the box was shifted to a neighbouring MA trimer, and the particles were re-extracted with a box size of 512 × 512 pixels and Fourier-cropped to 256 × 256 pixels. Duplicate particles were then removed. Lattice expansion increased the size of the dataset, improving the resolution of our reconstruction and the quality of the map. Two iterations of the lattice expansion were performed. To identify particles containing well-ordered MA lattice, the accepted particles (819,672 particles) were subjected to two rounds of 3D classification without angular search, using 10 classes. Particles were lattice-expanded as described above between the two 3D classifications. Classes showing a well-resolved MA layer were selected (174,245 particles) and subjected to a final round of local refinement, imposing $C_3$ symmetry and a mask comprising the MA layer resulting in a final map of MA with a resolution of 5.8 Å.

## Cryo-EM data processing for MA–SP1

The data processing pipeline for MA of MA–SP1 is summarized in Extended Data Fig. 4. A total of 14,222 motion-corrected micrographs and 5,704,512 initial particle positions were imported into cryoSPARC v3.3 (ref. 48). Particles, which were picked from large numbers of arbitrary positions over the virus surface, were extracted with a box size of 512 × 512 pixels and Fourier-cropped to 192 × 192 pixels. The first step was to perform a high-resolution reconstruction of the well-ordered immature CA layer to generate CA positions and orientations for use as priors to initialize the refinement of the MA layer. Particles were subjected to two rounds of 2D classification, and classes showing side and top views of the immature CA layer were selected (2,969,039 particles; Extended Data Fig. 4b). To accelerate computation, the selected particles were divided into 4 approximately equally sized subsets (each containing about 750,000 particles). Each subset underwent heterogeneous refinement with six classes, enforcing $C_6$ symmetry, using a previously determined structure of an in vitro-assembled HIV-1 capsid (EMDB accession code EMD-3782) as a starting reference 50 (Extended Data Fig. 4c). The highest-quality classes from each batch were then pooled for a series of refinement steps. They were subjected to non-uniform 3D refinement before particle re-extraction with a box size of 512 × 512 pixels, and Fourier-cropped to 384 × 384 pixels. Duplicate particles were removed on the basis of a spatial separation distance, and the accepted particles (885,809 particles) were subjected to 3D refinement with local angular and spatial searches (local refinement), imposed $C_6$ symmetry and a mask comprising the immature CA layer. The particles were then subjected to local CTF refinement, followed again by 3D refinement. Particles were re-extracted with a box size of 512 × 512 pixels, and Fourier-cropped to 450 × 450 pixels, and again subjected to local CTF refinement and local 3D refinement. Heterogeneous refinement was then performed to remove any remaining low-quality particles, using three classes, from which the highest-quality class was selected. Global CTF refinement[53] and further local refinement were then performed. Afterwards, the particles were imported to RELION-4.0 and subjected to Bayesian polishing[51]. The polished particles were imported back to cryoSPARC v3.3 and subjected to further 3D refinement and local CTF refinement to generate a final high-resolution immature CA reconstruction (Extended Data Fig. 4d).

The coordinates of the CA layer were then used to define initial coordinates and orientations for the MA-layer-focused reconstruction. The 3D coordinates of the centre of the box were shifted to the centre of the three-fold symmetry axis of the MA layer, which could be seen in the high-resolution CA reconstruction. Local refinement, without provision of a new reference, was then performed with imposed $C_3$

symmetry and a mask comprising only the MA layer (Extended Data Fig. 4e). Next, the dataset was expanded by using the refined positions of MA trimers to predict the positions of neighbouring MA trimers (lattice expansion). To do this, the particles were symmetry-expanded with $C_3$ symmetry (generating the two additional symmetry-related copies of each particle), and the centre of the box was shifted to the neighbouring six-fold symmetry axis, before duplicate particles were removed on the basis of a spatial separation distance. The accepted particles (1,398,844 particles) were subjected to a further round of local refinement with $C_6$ symmetry enforced. The particles were once again lattice-expanded, with the centre of the box shifted to the three-fold axis of neighbouring MA timers. Duplicate particles were removed, and the accepted particles (5,486,693 particles) were reconstructed with $C_3$ symmetry imposed to generate the final MA reconstruction (Extended Data Fig. 4e).

## Cryo-EM data processing for MA–NC

The processing strategy for MA–NC was almost identical to that of MA–SP1, including for the initial CA reconstruction and for the subsequent MA refinement steps. The data processing pipeline is summarized in Extended Data Fig. 5.

## Cryo-EM data processing for WT

The data processing pipeline for WT MA is summarized in Extended Data Fig. 6. A total of 19,530 motion-corrected micrographs and 5,801,053 initial particle positions were imported into cryoSPARC v3.3 (ref. 48). In contrast to the case for the PR⁻, MA–SP1 and MA–NC samples described above, there is no immature CA layer present in WT virions, so MA was reconstructed directly. Particles, which were picked from large numbers of arbitrary positions over the virus surface, were initially extracted with a box size of 480 × 480 pixels, Fourier-cropped to 240 × 240 pixels and subjected to 2 rounds of 2D classification to identify and select classes showing top and side views of the MA layer (Extended Data Fig. 6b). They were then subjected to two rounds of heterogeneous refinement, each with four classes and imposed $C_6$ symmetry, using a cryo-ET-derived mature MA lattice reconstruction as a starting reference (EMDB accession code EMD-13088; Extended Data Fig. 6c). The resulting highest-quality class was selected (61,672 particles) and subjected to non-uniform 3D refinement with $C_6$ symmetry imposed[52]. Duplicate particles were removed on the basis of spatial separation distance, and the accepted particles (57,260 particles) were then subjected to 3D refinement with local spatial and angular searches (local refinement), using a refinement mask comprised of the MA layer. The dataset was expanded by using the refined positions of MA trimers to predict the positions of neighbouring MA trimers (lattice expansion). To do this, the particles were symmetry-expanded with $C_3$ symmetry (generating the two additional symmetry-related copies of each particle), the centre of the box was shifted to the neighbouring six-fold symmetry axis, and the particles were re-extracted with a box size of 480 × 480 pixels and Fourier-cropped to 356 × 356 pixels (276,547 particles). The particles were subjected to local CTF refinement, followed by a further round of local refinement. The particles were once again lattice-expanded, with the centre of the box shifted to the neighbouring MA timers. Duplicates were again removed, and the accepted particles (898,502 particles) were finally reconstructed with imposed $C_3$ symmetry (Extended Data Fig. 6d).

## Automated lipid fitting

Automated docking of lipid candidates into the MA–SP1 ligand density (cholesterol, PtdIns(4,5)P₂, phosphatidylserine, phosphatidylcholine and phosphatidylethanolamine) was performed using RosettaEmerald, using the protocol described in ref. 54. All resulting fits were visually inspected and $Q$-scores of all ligand fits were determined using MapQ in USCF Chimera 1.15 (ref. 55). The top ten fits for each ligand, according to $Q$-score, are provided (Extended Data Fig. 7a).

## Fourier analysis of 2D class averages of cleavage mutants

Cryo-EM data preprocessing and particle picking were performed as described in the cryo-EM subsection. Images were extracted with a box size of 480 × 480 pixels and downsampled to 240 × 240 pixels resulting in a final pixel size of 1.9 Å. 2D class averages of mutants that displayed side views of membranes were selected manually in cryoSPARC v4.4. Afterwards, the 2D classes were reoriented according to a reference class in which the membrane bilayer was oriented perpendicularly to the $y$ axis of the 2D class using cross-correlation (Extended Data Fig. 8). Then, the pixel values along the MA layer (section parallel to the inner leaflet of the viral membrane) were interpolated with 1,024 points and exported as 1D vectors. The vector was filtered using a Hann function and zero-padded to a total of 4,096 sampling points. Fast Fourier transformation of the zero-padded signal was plotted in MATLAB v2022a (MathWorks) and analysed for peaks corresponding to MA lattice spacing frequencies. All of the signal processing steps were performed in MATLAB v2022a (MathWorks).

## Atomic model building and refinement of SP2 bound to MA

The solution structure of myristoylated HIV-1 MA (myrMA; Protein Data Bank accession code 2H3I)[12] was used as an initial MA structure for building into the MA–SP1 density. Initial coordinates for $SP2_{11-16}$ were generated using AlphaFold (v2.2.0)[56], which were fitted roughly into the SP2 density in USCF Chimera[55]. The initial model was then flexibly fitted, with manual adjustments, into the density with ISOLDE 1.3 (in ChimeraX 1.3)[57,58]. A single round of real-space refinement was then performed in Phenix-1.21. Final model validation statistics and the map-to-model Fourier shell correlation were calculated in Phenix-1.21 (ref. 59) and are given in Extended Data Table 2.

## ModelAngelo predictions of SP2

The 3.1-Å-resolution map of the mature MA of the MA–SP1 cleavage mutant was used for automated ModelAngelo v1.0 (ref. 60) atomic model predictions. The box size was cropped to 128 × 128 pixels and only sequences built into the central trimer were considered. The ModelAngelo job was run with default parameters in RELION-5.0. The sequence input was either the HIV-1 Gag sequence or no sequence. Afterwards, sequences built into the side pocket cryo-EM densities from the central MA trimer were extracted and analysed by the Clustal Omega multiple sequence analysis tool[61]. The ModelAngelo automated model building and sequence prediction results are shown in Extended Data Fig. 7b–d.

## Expression and purification of HIV-1 MA

The expression plasmid pET11b-MA encodes the HIV-1 pNL4−3 MA domain with a six-residue C-terminal His tag[62]. BL21(DE3) *Escherichia coli* competent cells for protein expression were co-transformed with the pET11b-MA plasmid and a plasmid encoding the yeast N-terminal myristoyltransferase. The protein was expressed and purified as described previously[17,63] with modifications. Cells were grown at 37 °C at 180 rpm in a lysogeny broth medium containing 100 mg l⁻¹ of ampicillin and 50 mg l⁻¹ of kanamycin. When absorbance at 600 nm ($A_{600nm}$) reached about 0.6, 15 mg l⁻¹ myristic acid (Sigma-Aldrich) was added to the lysogeny broth medium. After 30 min, cells were induced with 0.5 mM isopropyl β-D-1-thiogalactopyranoside and grown for another 5 h at 37 °C at 180 rpm. Afterwards, the cells were spun down at 3,500$g$, and the pellets were stored at −80 °C until further use. For purification, 6 g of the cell pellet was diluted in 60 ml of lysis buffer (25 mM Tris pH 8, 500 mM NaCl, 2 mM TCEP and 2 mM phenylmethylsulfonyl fluoride) and sonicated for 2 min. Then, 20 µl of benzonase was added to the lysed cells. The lysate was incubated on ice for 10 min and then spun down at 50,000 rpm at 10 °C for 45 min (Beckman Coulter, 50.2 TI rotor). The supernatant was collected and treated

with polyethyleneimine to a final concentration of 0.03%, incubated on ice for 5 min and subsequently centrifuged at 10,000 rpm at 4 °C for 10 min (Beckman Coulter, JA-25.50 rotor). The supernatant was collected, and powdered ammonium sulfate (approximately 15 g) was added to the supernatant on ice with constant stirring until protein precipitate was observed, followed by centrifugation at 10,000 rpm at 4 °C for 10 min (Beckman Coulter, JA-25.50 rotor). The pellet was resuspended in 4 ml of binding buffer (20 mM Tris-HCl, pH 8, 100 mM NaCl, 2 mM TCEP) and loaded onto a HisTrap column (Cytiva) equilibrated with the binding buffer. The column was then washed with a wash buffer (20 mM Tris-HCl, pH 8, 100 mM NaCl, 2 mM TCEP and 20 mM imidazole), and the protein was subsequently eluted with elution buffer (20 mM Tris-HCl, pH 8, 100 mM NaCl, 2 mM TCEP and 250 mM imidazole). Fractions containing myrMA were collected and further purified by gel filtration using Superdex 75 16/600 (Cytiva) equilibrated in buffer containing (20 mM Tris-HCl, pH 8, 500 mM NaCl, 1 mM TCEP). Fractions containing myrMA were collected and stored at −80 °C. The presence of myristoylation modification was confirmed by mass spectrometry.

## 2D crystallization of HIV MA

2D crystallizations were performed in a cleaned polytetrafluoroethylene block containing 60-ml side-entry reservoirs, combining previous protocols[64]. A 58 ml volume of crystallization buffer (12 mM sodium phosphate buffer pH 7.8; 2.5 mM sodium acetate pH 7.6; 150 mM sodium chloride; 10% glycerol)[33,65] was added to 6 crystallization reservoirs. Afterwards, 1 µl of a freshly prepared lipid mixture mimicking the inner leaflet of the viral membrane (molar fractions: 31% cholesterol; 6% POPC; 29% POPE; 27% POPS; 7% PtdIns(4,5)$P_2$ (ref. 34)) in 9:1 (v/v) chloroform/methanol solution at a lipid concentration of 0.01 mg ml⁻¹ was carefully added on top of each buffer surface. The polytetrafluoroethylene block was then incubated in a closed Petri dish with a wet filter paper placed underneath the block for 60 min to allow a lipid monolayer to form at the air–water interface. Then a Quantifoil 2/2 holey carbon grid, Au 200 mesh (Quantifoil Micro Tools) was placed on top of each reservoir. A solution containing purified myrMA was then injected into each reservoir from the side entrance. The final concentration of myrMA in the reservoir was 12 mM. After 10 min, HIV-1 SP2 in PBS buffer was added to 3 of the 6 experimental reservoirs to a final concentration of 120 mM. The same amount of PBS buffer was added to the three control reservoirs. All samples were incubated for an additional 60 min, and then the grids were carefully lifted from the surface of the reservoirs and plunge-frozen using the Vitrobot Mark IV (4 s blot time; 3 blot force; 100% humidity).

Grids were loaded into a Talos Glacios cryo-transmission electron microscope operating at 200 kV and equipped with a Falcon4i direct electron detector (ThermoFisher Scientific). For each grid, 5 grid squares were selected manually and in each square 30 holes were randomly selected in the EPU software (v3). A single acquisition position was selected in the centre of each hole resulting in 150 micrographs automatically collected from each grid. The micrographs were collected as videos of 40 frames with a total dose of 40 e⁻ Å⁻² at a magnification of ×92,000 resulting in a pixel size of 1.20 Å per pixel. Videos were motion-corrected, dose-weighted and averaged using the RELION-4.0 MotionCorr2 algorithm[46,47]. CTF estimation was performed using CTFfind4[66]. Particles were picked as a grid of points separated by 128 pixels placed in a 4,000 × 4,000 micrograph, resulting in 139,500 particles for each dataset (Extended Data Fig. 11b). Particles were extracted with a box size of 256 pixels and Fourier-cropped to 128 pixels. The particles were then imported to cryoSPARC v4.4 (ref. 48) and subjected to two rounds of 2D classification (Extended Data Fig. 11). Classes showing a 2D crystal MA lattice were selected, and particles from these classes were considered as particles containing a 2D crystal lattice. All six datasets were processed the same way. To facilitate visual

comparison of class averages and simulations in Fig. 4e–h, greyscales were made similar using Fiji (ImageJ v1.54f).

## Plasmids, reagents and cell lines for fusion assays

HIV-1 GagPol was expressed by pCMV ΔR8.2 (Addgene plasmid no. 12263). The pCAGGS HIV-1$_{JRFL}$ gp160 expression plasmid was provided by J. Binley. The pN1 CypA–HiBiT plasmid (made by J. Grover) was derived from pEGFP-N1-CypA. To generate this plasmid, human cyclophilin A was cloned from HeLa cDNA and inserted into pEGFP-N1 using the EcoRI and BamHI sites. To generate CypA–HiBiT, a synthetic oligonucleotide, which encoded the linker sequence GSGSSGGGGSGGGGSSG followed by the HiBiT peptide VSGWRLFKKIS, was inserted to replace eGFP at the C terminus of CypA using the BamHI and NotI sites. This construct is based on a similar design by G. Melikyan.

The Gag cleavage mutants were constructed through site-directed mutagenesis and Gibson assembly. The alterations for pCMV ΔR8.2 MA–CA, pCMV ΔR8.2 MA–SP1, pCMV ΔR8.2 MA–NC and pCMV ΔR8.2 MA–p6 were recreated from previous literature[6,67,68]. Gene blocks of the MA–CA, MA–SP1, MA–NC and MA–p6 GagPol sequences were ordered from Twist Bioscience and combined with fragments of the pCMV ΔR8.2 backbone amplified by PCR for Gibson assembly using the Gibson Assembly Master Mix from New England Biolabs. The alterations for pCMV ΔR8.2 MA–SP2 and pCMV ΔR8.2 NC–p6 were created using site-directed mutagenesis using the pCMV ΔR8.2 MA–p6 and pCMV ΔR8.2 constructs as templates, respectively. These fragments were combined with pCMV ΔR8.2 backbone fragments amplified by PCR and combined with Gibson assembly using the same protocol as above.

The human CD4-expressing vector pcDNA-hCD4 was provided by H. Gottlinger. The pMX-puro PH-PLC∂LgBiT plasmid was provided by Z. Matsuda[30]. The following reagents were obtained through the National Institutes of Health (NIH) HIV Reagent Program, Division of AIDS, National Institute of Allergy and Infectious Diseases (NIAID), NIH: indinavir sulfate, ARP-8145, and ritonavir, ARP-4622, both contributed by Division of AIDS, NIAID; human CCR5 expression vector (pcCCR5), ARP-3325, contributed by N. Landau.

HEK293 cells (ATCC no. CRL-1573) were grown in the presence of 5% $CO_2$ using RPMI-1640 medium from ThermoFisher Scientific supplemented with 10% fetal bovine serum (Invitrogen), 100 U ml$^{-1}$ of a penicillin and streptomycin solution and 2 mM L-glutamine. Cells were transfected at 60%–80% confluency, and culture medium was exchanged before transfection. Cells were regularly tested negative for mycoplasma contamination.

## Virus-like particle preparation for fusion kinetics

Virus-like particles (VLPs) were produced by transfecting three 10-cm plates of HEK293 cells with 12 µg DNA per 10-cm plate using polycation polyethylenimine (pH 7.0, 1 mg ml$^{-1}$). Immature VLP preparations were transfected with a final concentration of 1 µM indinavir and 10 µM ritonavir in the cell culture medium. Plasmids were transfected in a 1:1:1 ratio of Gag/Env/CypA-HiBiT. Cell culture medium was collected 2 days after transfection, and then spun down for 5 min to pellet cells. Supernatants were transferred to 38.5-ml ultracentrifuge tubes and underlaid with 5 ml sterile-filtered 15% sucrose in PBS. VLPs were then pelleted by ultracentrifugation at a maximum of 131,453 RCF using a Beckman Coulter SW28 swinging bucket rotor at 27,000 rpm for 1 h at 4 °C. Supernatant was removed and viruses were resuspended in 1:100 volume (300 µl) of serum-free $CO_2$-independent medium (ThermoFisher Scientific). Particles produced in the presence of protease inhibitors were resuspended in serum-free $CO_2$-independent medium with final concentrations of 1 µM indinavir and 10 µM ritonavir. Each VLP preparation was aliquoted into about 20 tubes at 15 µl per tube and stored at −80 °C until use. After collection, each aliquot of VLPs was used once for fusion kinetics assays to minimize particle destruction during freeze–thaw cycles.

## VLP normalization for fusion kinetics

VLP volumes were normalized on HiBiT incorporation using the Nano-Glo HiBiT Lytic Detection System from Promega according to the manufacturer's instructions. Three volumes of the VLPs (2 µl, 4 µl and 6 µl) were measured per sample. Plates were placed in a Promega GloMax Explorer GM3500 Multimode Microplate Reader and read using the manufacturer's suggested protocol. Readings were then graphed in Excel with a linear trendline. The trendline equation was used to calculate the volume of each sample containing $3 × 10^7$ relative light units of HiBiT.

## Fusion kinetics assay

The split nanoluciferase fusion kinetics assay described in ref. 30 was modified to enable real-time live monitoring of fusion events. Briefly, HEK293 cells were transfected with a 1:1:1 ratio of CD4, CCR5 and PH-LgBiT. After 24 h, cells were collected and Endurazine (Promega) and DrkBiT peptide were added to the solution to a final concentration of 1× and 1 µg ml$^{-1}$, respectively. A white 96-well flat-bottom plate was prepared with 100 µl of the cell solution at a density of about $2 × 10^4$ cells per well. Prepared VLPs were added to the wells and spinoculation was performed at 1,200 RCF and 12 °C for 2 h. The plate was then covered with sterile BREATH-EASY*GAS PERMEABLE film (USA Scientific, no. 9123-6100). The plate was read continuously in a Promega GloMax Explorer GM3500 Multimode Microplate Reader using an automatic protocol for 24 rounds of reading wells every 2 min with a 1.5 s integration time per well at 37 °C. Readings were exported into Excel, in which relative light units were calculated, and then processed into percentage of total fusion. The percentage of total fusion curves were processed using an in-house Mathematica code (Supplementary Data 1; written by A. Lee) to generate $T_{1/2}$ for each sample. Results were plotted using GraphPad Prism v10.4.1.

## Reporting summary

Further information on research design is available in the Nature Portfolio Reporting Summary linked to this article.

## Data availability

Structures determined by electron microscopy have been deposited in the EMDB under the accession codes EMD-52229, EMD-51769, EMD-52221 and EMD-52222. The molecular model of the MA lattice of MA–SP1 has been deposited in the Protein Data Bank under the accession code 9H1P, for which 2H3I was used as a starting model.

## Code availability

Custom code used in the analysis of viral fusion assays is provided in Supplementary Data 1.

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

**Acknowledgements** This work was financed by the Deutsche Forschungsgemeinschaft (DFG, German Research Foundation) Projektnummer 240245660 – SFB 1129 (project 5, H.-G.K.; project 6, B.M.; project 21, J.A.G.B.), DFG KR 906/7-1 (H.-G.K.), NIH NIAID R37 AI150560 (W.M.) and the Max Planck Society (J.A.G.B.). E.N. was supported by NIH NIAID T32 AI055403. Cryo-EM data were collected in the Department of Cell and Virus Structure, Max Planck Institute of Biochemistry. We thank L. Regner for plasmid pcHIV(MA–SP1:NC–p6); H. Guo and Z. Ke for assistance with data collection; Z. Ke for advice on data analysis; F. Beck, J. R. Prabu and I. Wolf for assistance with computing infrastructure; and A. Lee for assistance with fusion kinetics data analysis.

**Author contributions** J.C.V.S., D.H., E.N., P.D.U., W.M., B.M., H.-G.K. and J.A.G.B. designed research; S.D.S. and M.A.-O. prepared and analysed virus samples; K.Q. performed preliminary experiments. M.B. prepared protein; R.A.D. provided reagents and advice. J.C.V.S. and D.H. carried out cryo-EM and related data processing; J.C.V.S., D.H. and J.A.G.B. analysed and interpreted structural data; E.N. performed fusion kinetics experiments and analysis; J.C.V.S., D.H., E.N., W.M. and J.A.G.B. designed and prepared figures; J.C.V.S., D.H. and J.A.G.B. wrote the manuscript with input from all authors; B.M., H.-G.K., W.M. and J.A.G.B. obtained funding and managed the project.

**Funding** Open access funding provided by Max Planck Society.

**Competing interests** The authors declare no competing interests.

**Additional information**
**Correspondence and requests for materials** should be addressed to John A. G. Briggs.

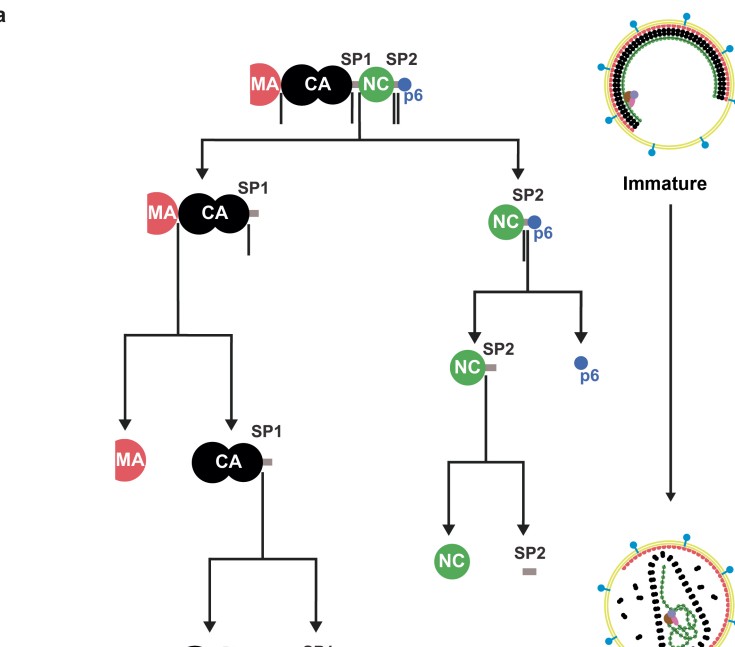

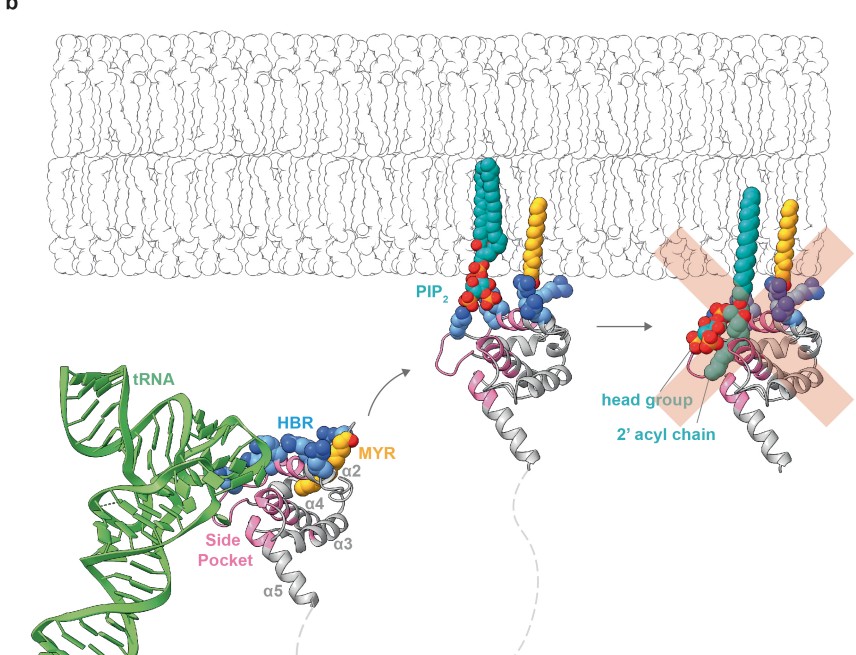

**Extended Data Fig. 1 | Schematic of the Gag cleavage cascade and current model of MA interactions at the plasma/viral membrane. a**, Schematic illustration of the Gag proteolytic cleavages that occur during HIV-1 maturation. Cleavages are ordered by relative rate as measured in-vitro[69]. Virus schematics adapted from Briggs and Kräusslich (2011)[70] **b**, MA in the cytoplasm can bind tRNA (green) via the side pocket. MA binds to PtdIns(4,5)P$_2$ (PIP$_2$) (blue lipid) and cholesterol rich domains on the plasma membrane during viral assembly. This binding is facilitated by the insertion of a previously sequestered myristoyl group (yellow) and by electrostatic interactions at the HBR. Bound tRNA sterically prevents membrane binding of MA, thus tRNA$^{Lys3}$ is released upon membrane binding, leaving the side pocket unoccupied. The current model, based on data from NMR[12] and our previous cryo-ET study[1], is that that upon maturation, MA partially extracts molecules of PtdIns(4,5)P$_2$ from the inner leaflet of the viral membrane, binding the phosphatidylinositol head group and one of the acyl tails within the side pocket. This aspect of the model is, however, contradicted by the findings of this study (indicated by red cross).

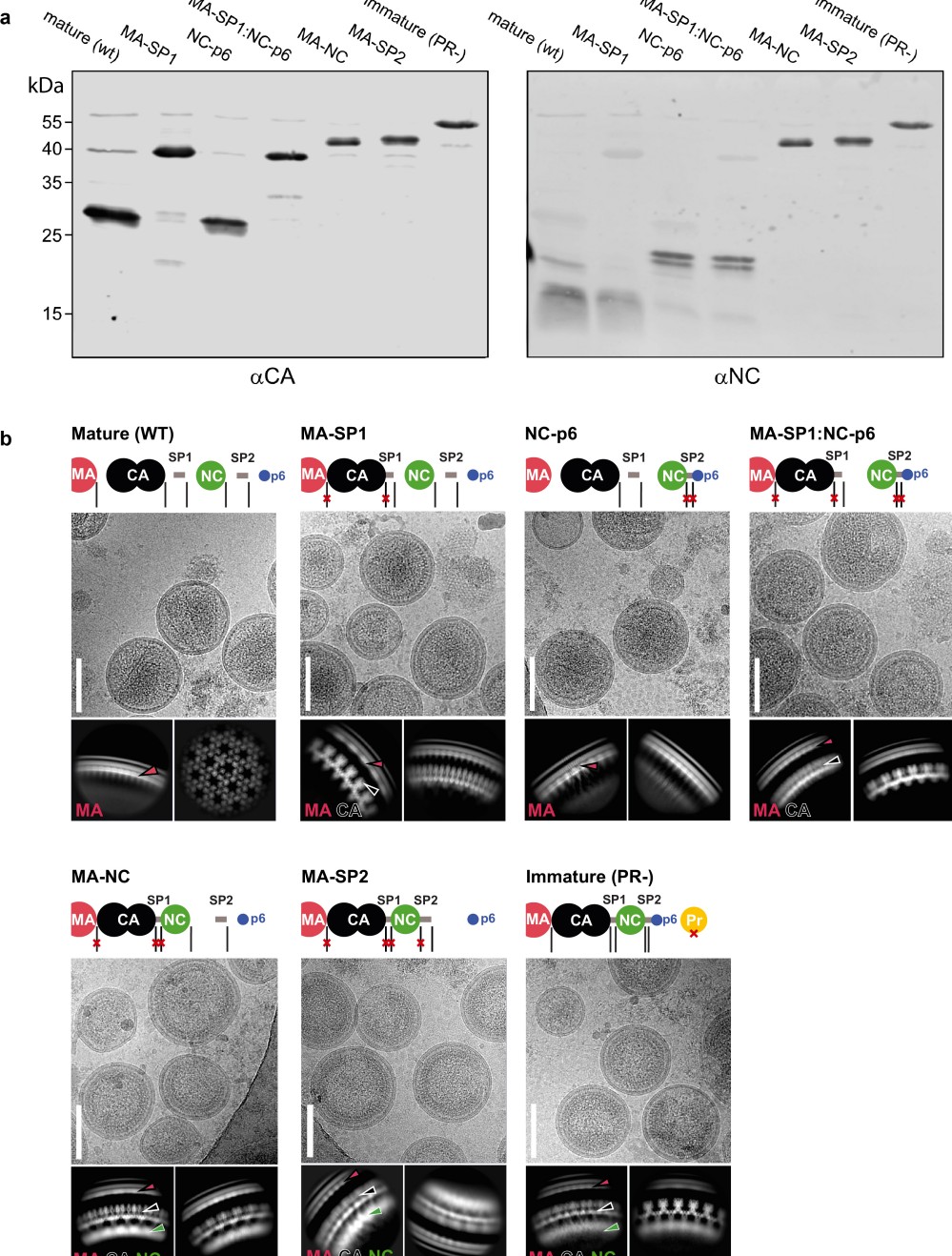

**Extended Data Fig. 2 | Representative micrographs, 2D classes and western blot analysis of HIV-1 particles used in this study. a**, HEK293T cells were transfected with pcHIV carrying the indicated Gag cleavage site mutations. Particles were collected at 44–48 h post transfection by ultracentrifugation. Viral proteins were detected by immunoblotting with the indicated polyclonal antisera. Observed cleavage patterns are representative of at least three independent preparations for each variant. Antibodies were detected using a LI-COR Odyssey Clx infrared scanner, using secondary antibodies and protocols provided by the instrument's manufacturer. Positions of molecular mass markers in kDa are indicated at the left. For gel source data, see Supplementary Fig. 1. **b**, For each dataset a representative cryo-EM micrograph (from at least 1000 micrographs of virus) is shown, illustrating the overall viral morphology (scale bars: 100 nm). Representative high resolution 2D classes of the Gag layer are also shown, labelled with the features that could be identified, including MA, CA. and NC layers. Schematic illustrations of the processing state of Gag constructs are given for each dataset.

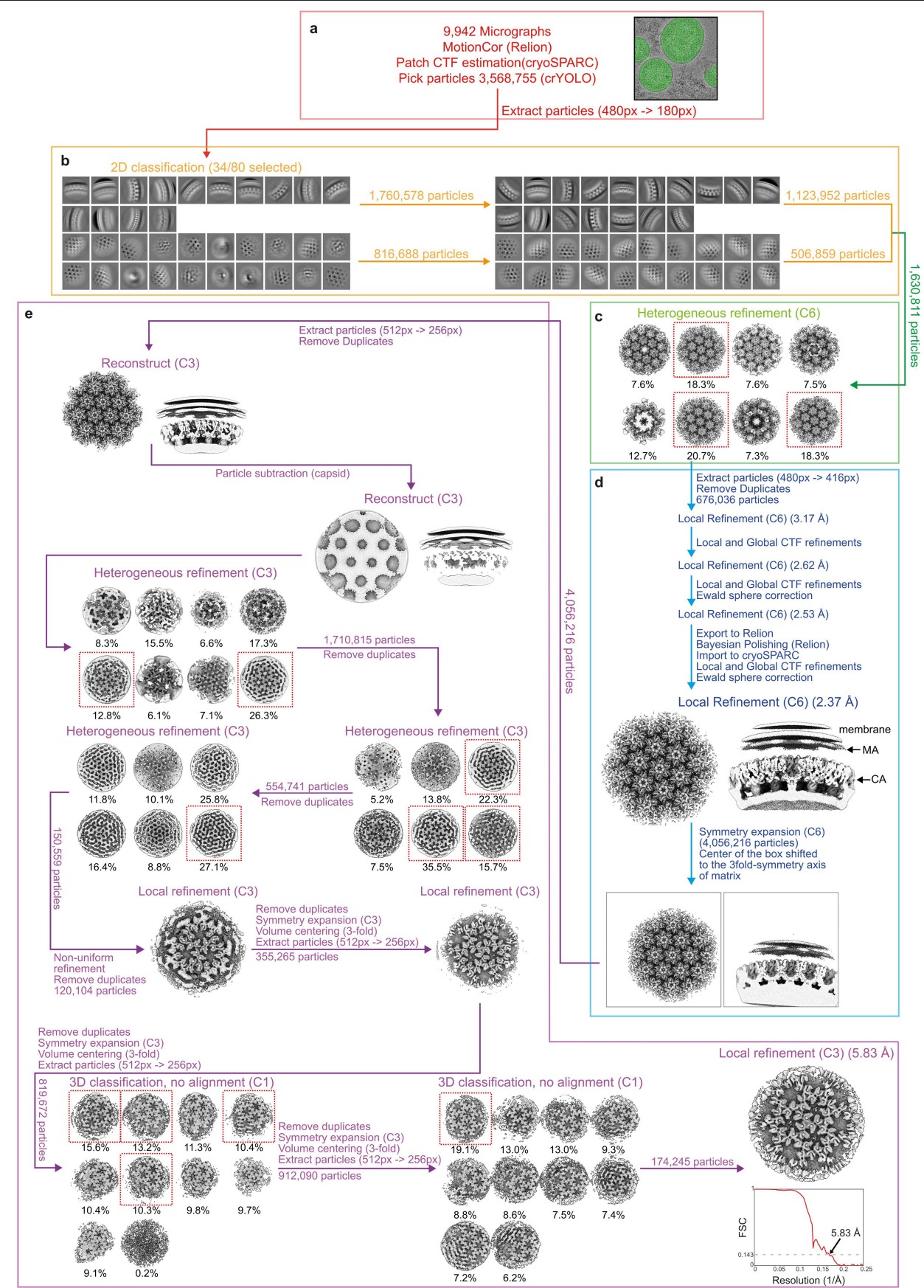

**Extended Data Fig. 3** | See next page for caption.

**Extended Data Fig. 3 | Single-particle cryo-EM processing pipeline for the Immature (PR-) MA lattice.** PR- virus particles consist of an uncleaved Gag polyprotein. The MA lattice is in an immature configuration. **a**, EM micrograph pre-processing, particle picking and particle extraction. Micrographs were motion corrected using MotionCor2 in Relion and CTF estimation was performed using patch CTF estimation in cryoSPARC. Picking was performed using crYOLO, where a model was trained to pick arbitrary positions across the entire virus surface. **b**, Two rounds of 2D classification were used to identify high quality particles containing the immature CA layer. After the first round, the particles were separated into two groups - top and side views. In the second 2D classification, these two groups were classified independently. The best classes were selected and passed to heterogeneous refinement. **c**, Heterogeneous refinement was performed to identify the highest-quality immature CA particles, which were selected for further processing. **d**, Several rounds of CTF refinements, local defocus refinements, Ewald sphere corrections, local refinements and one iteration of Bayesian polishing in Relion 4.0 were performed to obtain a high-resolution immature CA map (2.37 Å). Within the resulting reconstruction, diffuse but recognisable density corresponding to the immature MA layer above CA was observed. **e**, Refinement of the immature MA layer. To generate a reconstruction focused on the MA layer, the CA particles were symmetry expanded and the centre of the box was first shifted from the immature CA lattice to the three-fold symmetry centre of the diffuse MA layer. Particles were re-extracted with the particle centred on the MA layer while the strongly featured immature CA layer was subtracted. Then, three iterations of heterogeneous refinements of the MA lattice were performed to identify particles with ordered MA lattice, followed by non-uniform refinement. The fact that MA forms a 2D lattice was then exploited to add particles at the expected positions of symmetry related neighbours, thereby expanding the set of particles (lattice expansion). At this stage, the particles are centred on the three-fold symmetry axis. To perform lattice expansion, for each particle, three symmetry-related particles were created by symmetry expansion and were assigned positions and angles corresponding to the neighbouring three-fold symmetry axes (as determined from the 3D reconstruction), followed by centering of the box to the neighbouring 3-fold symmetry axes of MA (lattice expansion). We found that this improved the resolution of our reconstruction and the quality of the map. That was followed by another lattice expansion and by 3D classification without angular search, followed by another round of lattice expansion and a final 3D classification. The best-looking class showing secondary structure features was selected and was refined using a local refinement with imposed C3 symmetry, resulting in a map of immature MA lattice with a resolution of 5.83 Å.

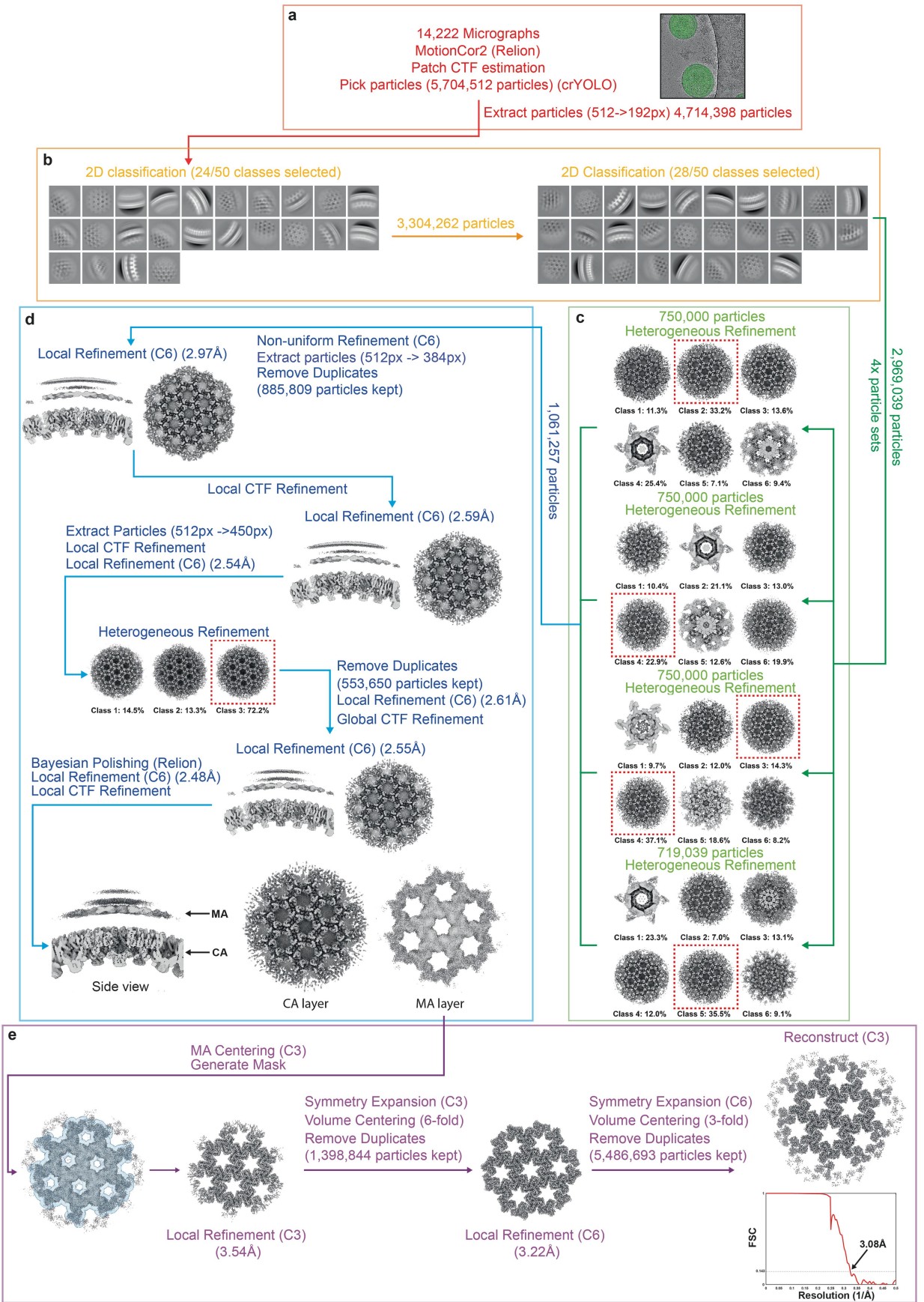

**a**
14,222 Micrographs
MotionCor2 (Relion)
Patch CTF estimation
Pick particles (5,704,512 particles) (crYOLO)

Extract particles (512->192px) 4,714,398 particles

**b**
2D classification (24/50 classes selected)

2D Classification (28/50 classes selected)

3,304,262 particles

**c**
750,000 particles
Heterogeneous Refinement

Class 1: 11.3%    Class 2: 33.2%    Class 3: 13.6%
Class 4: 25.4%    Class 5: 7.1%     Class 6: 9.4%

750,000 particles
Heterogeneous Refinement

Class 1: 10.4%    Class 2: 21.1%    Class 3: 13.0%
Class 4: 22.9%    Class 5: 12.6%    Class 6: 19.9%

750,000 particles
Heterogeneous Refinement

Class 1: 9.7%     Class 2: 12.0%    Class 3: 14.3%
Class 4: 37.1%    Class 5: 18.6%    Class 6: 8.2%

719,039 particles
Heterogeneous Refinement

Class 1: 23.3%    Class 2: 7.0%     Class 3: 13.1%
Class 4: 12.0%    Class 5: 35.5%    Class 6: 9.1%

2,969,039 particles
4x particle sets

1,061,257 particles

**d**
Local Refinement (C6) (2.97Å)

Non-uniform Refinement (C6)
Extract particles (512px -> 384px)
Remove Duplicates
(885,809 particles kept)

Local CTF Refinement

Local Refinement (C6) (2.59Å)

Extract Particles (512px ->450px)
Local CTF Refinement
Local Refinement (C6) (2.54Å)

Heterogeneous Refinement

Class 1: 14.5%    Class 2: 13.3%    Class 3: 72.2%

Remove Duplicates
(553,650 particles kept)
Local Refinement (C6) (2.61Å)
Global CTF Refinement

Local Refinement (C6) (2.55Å)

Bayesian Polishing (Relion)
Local Refinement (C6) (2.48Å)
Local CTF Refinement

MA
CA

Side view        CA layer        MA layer

**e**
MA Centering (C3)
Generate Mask

Reconstruct (C3)

Symmetry Expansion (C3)
Volume Centering (6-fold)
Remove Duplicates
(1,398,844 particles kept)

Symmetry Expansion (C6)
Volume Centering (3-fold)
Remove Duplicates
(5,486,693 particles kept)

Local Refinement (C3)
(3.54Å)

Local Refinement (C6)
(3.22Å)

3.08Å

FSC

Resolution (1/Å)

**Extended Data Fig. 4** | See next page for caption.

**Extended Data Fig. 4 | Single-particle cryo-EM processing for MA-SP1 MA lattice.** MA-SP1 virions consist of a partially cleaved Gag layer, where the immature CA layer is intact but the MA lattice is in the mature conformation[1]. **a**, EM micrograph pre-processing, particle picking and particle extraction. Micrographs were motion corrected using MotionCor2 in Relion and CTF estimation was performed using patch CTF estimation in cryoSPARC. Picking was performed using crYOLO, where a model was trained to pick arbitrary positions across the entire virus surface. **b**, Two rounds of 2D classification were employed to identify and select high quality particles containing the immature CA layer. The immature CA layer was readily observed within top-view and side-view classes, however top view classes converged exclusively onto the strongly featured immature CA layer. **c**, Heterogenous refinement was performed in 3D, focused on the immature CA layer, to further differentiate between high quality and junk particles. To speed up computation here the dataset was split into four roughly equally sized subsets which were processed in parallel. **d**, High quality particles containing the immature CA layer, as determined by 2D classification and heterogenous refinement, were pooled and subjected to rounds of global 3D refinement, local 3D refinement, heterogenous refinement, local CTF refinement, global CTF refinement and Bayesian polishing, in order to generate a high-resolution (2.48 Å)

reconstruction of the immature CA layer. Within the resulting reconstruction, diffuse but recognisable density corresponding to the mature MA layer above CA was observed. **e**, Local refinement of the MA layer. To generate a reconstruction focused on the MA layer, the centre of the box was first shifted from the immature CA lattice to the centre of the diffuse MA layer. A mask was generated fitting the MA density layer which was then used in the local refinement of the MA layer, (without introduction of an additional reference). The subsequent refinement converged to a 3.54 Å, MA-focused, reconstruction. The fact that MA forms a 2D lattice was then exploited to add particles at the expected positions of symmetry related neighbours, thereby expanding the set of particles (lattice expansion). At this stage the particles are centred on the three-fold axis. To perform lattice expansion, for each particle, three symmetry-related particles were created and were assigned positions and angles corresponding to the neighbouring hexameric holes (as determined from the 3D reconstruction). The lattice expansion step was then repeated with six particles created and assigned positions and angles corresponding to the neighbouring trimers. At each step, any duplicate particles were removed. The expanded particle set was reconstructed to generate a final reconstruction of the mature MA lattice at a resolution 3.08 Å.

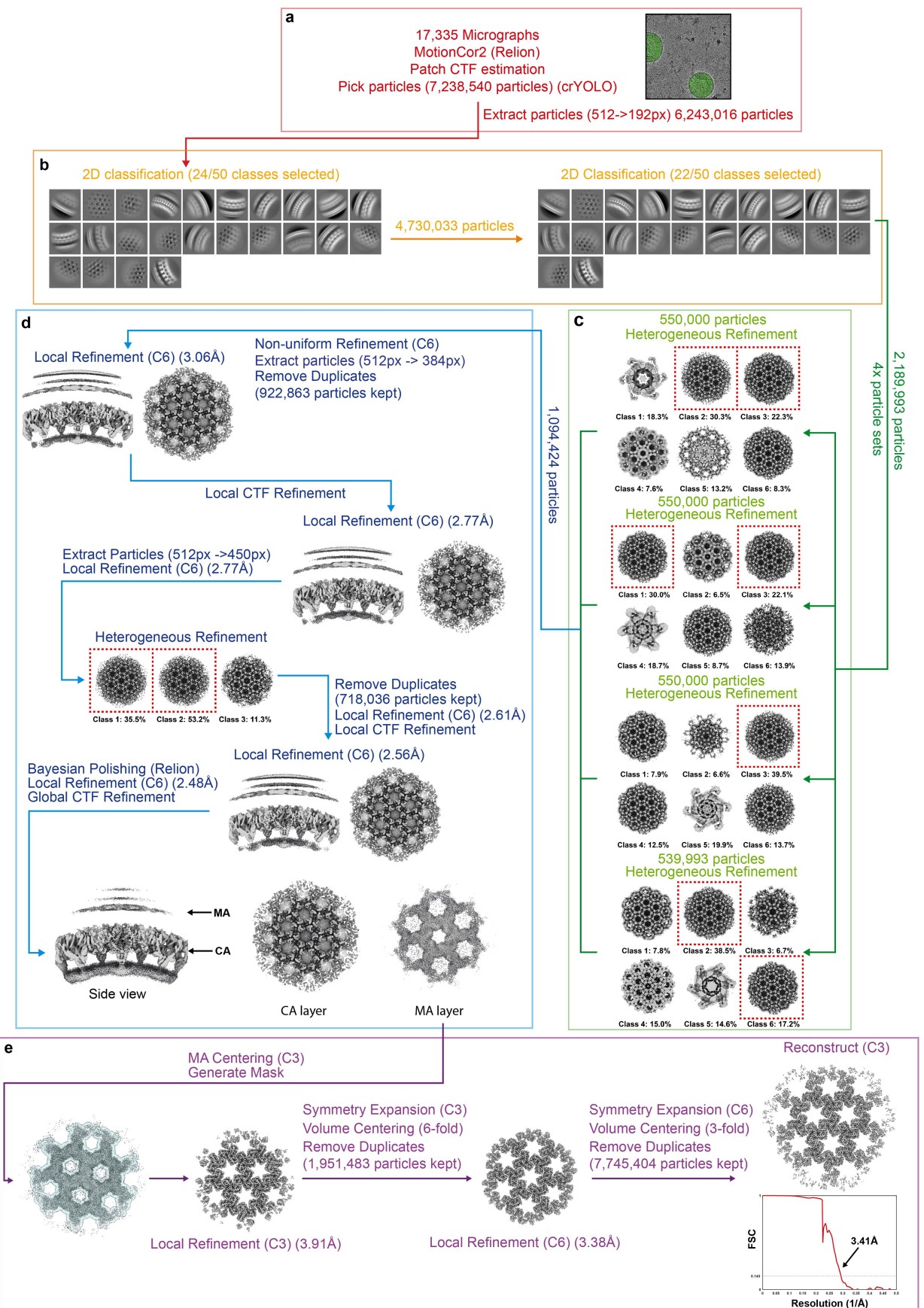

**a**

17,335 Micrographs
MotionCor2 (Relion)
Patch CTF estimation
Pick particles (7,238,540 particles) (crYOLO)

Extract particles (512->192px) 6,243,016 particles

**b**

2D classification (24/50 classes selected)          2D Classification (22/50 classes selected)

4,730,033 particles

**d**

Local Refinement (C6) (3.06Å)          Non-uniform Refinement (C6)
Extract particles (512px -> 384px)
Remove Duplicates
(922,863 particles kept)

Local CTF Refinement

Local Refinement (C6) (2.77Å)

Extract Particles (512px ->450px)
Local Refinement (C6) (2.77Å)

Heterogeneous Refinement

Remove Duplicates
(718,036 particles kept)
Local Refinement (C6) (2.61Å)
Local CTF Refinement

Class 1: 35.5%    Class 2: 53.2%    Class 3: 11.3%

Local Refinement (C6) (2.56Å)

Bayesian Polishing (Relion)
Local Refinement (C6) (2.48Å)
Global CTF Refinement

MA

CA

Side view          CA layer          MA layer

**c**

550,000 particles
Heterogeneous Refinement

Class 1: 18.3%    Class 2: 30.3%    Class 3: 22.3%

Class 4: 7.6%    Class 5: 13.2%    Class 6: 8.3%

550,000 particles
Heterogeneous Refinement

Class 1: 30.0%    Class 2: 6.5%    Class 3: 22.1%

Class 4: 18.7%    Class 5: 8.7%    Class 6: 13.9%

550,000 particles
Heterogeneous Refinement

Class 1: 7.9%    Class 2: 6.6%    Class 3: 39.5%

Class 4: 12.5%    Class 5: 19.9%    Class 6: 13.7%

539,993 particles
Heterogeneous Refinement

Class 1: 7.8%    Class 2: 38.5%    Class 3: 6.7%

Class 4: 15.0%    Class 5: 14.6%    Class 6: 17.2%

1,094,424 particles

2,189,993 particles

4x particle sets

**e**

MA Centering (C3)
Generate Mask

Reconstruct (C3)

Symmetry Expansion (C3)
Volume Centering (6-fold)
Remove Duplicates
(1,951,483 particles kept)

Symmetry Expansion (C6)
Volume Centering (3-fold)
Remove Duplicates
(7,745,404 particles kept)

Local Refinement (C3) (3.91Å)          Local Refinement (C6) (3.38Å)

FSC

3.41A

Resolution (1/A)

**Extended Data Fig. 5** | See next page for caption.

**Extended Data Fig. 5 | Single-particle cryo-EM processing for MA-NC MA lattice.** The processing pipeline for reconstruction of the MA-NC MA lattice is largely the same as that for MA-SP1. MA-NC virions consist of a partially cleaved Gag layer, where the immature CA layer is intact but the MA lattice is in the mature conformation (Fig. 2c). **a**, EM micrograph pre-processing, particle picking and particle extraction. Micrographs were motion corrected using MotionCor2 in Relion and CTF estimation was performed using patch CTF estimation in cryoSPARC. Picking was performed using crYOLO, where a model was trained to pick arbitrary positions across the entire virus surface. **b**, Two rounds of 2D classification were employed to identify and high quality particles containing the immature CA layer. The immature CA layer was readily observed within top-view and side-view classes, however top view classes converged exclusively onto the strongly featured immature CA layer. **c**, Heterogenous refinement was performed in 3D, focused on the immature CA layer, to further differentiate between high quality and junk particles. To speed up computation here the dataset was split into four roughly equally sized subsets which were processed in parallel. **d**, High quality particles containing the immature CA layer, as determined by 2D classification and heterogenous refinement, were pooled and subjected to rounds of global 3D refinement, local 3D refinement, heterogenous refinement, local CTF refinement, global CTF refinement and Bayesian polishing, in order to generate a high-resolution (2.48 Å) reconstruction of the immature CA layer. Within the resulting reconstruction diffuse but recognisable density corresponding to the mature MA layer above CA was observed. **e**, Local refinement of the MA layer. To generate a reconstruction focused on the MA layer, the centre of the box was first shifted upwards from the immature CA lattice to the centre of the diffuse MA layer. A mask was generated fitting the MA density layer which was then used in the local refinement of the MA layer (without introduction of an additional reference). The subsequent refinement converged to a 3.91 Å, MA focused, reconstruction. The fact that MA forms a 2D lattice was then exploited to add particles at the expected positions of symmetry related neighbours, thereby expanding the set of particles (lattice expansion). At this stage the particles were centred on the three-fold axis. To perform lattice-expansion, for each particle, three particles were created and were assigned positions and angles corresponding to the neighbouring hexameric holes (as determined from the 3D reconstruction). The lattice expansion step was then repeated with six particles created and assigned positions and angles corresponding to the neighbouring trimers. At each step, any duplicate particles were removed. The expanded particle set was reconstructed to generate a final reconstruction of the mature MA lattice at a resolution of 3.41 Å.

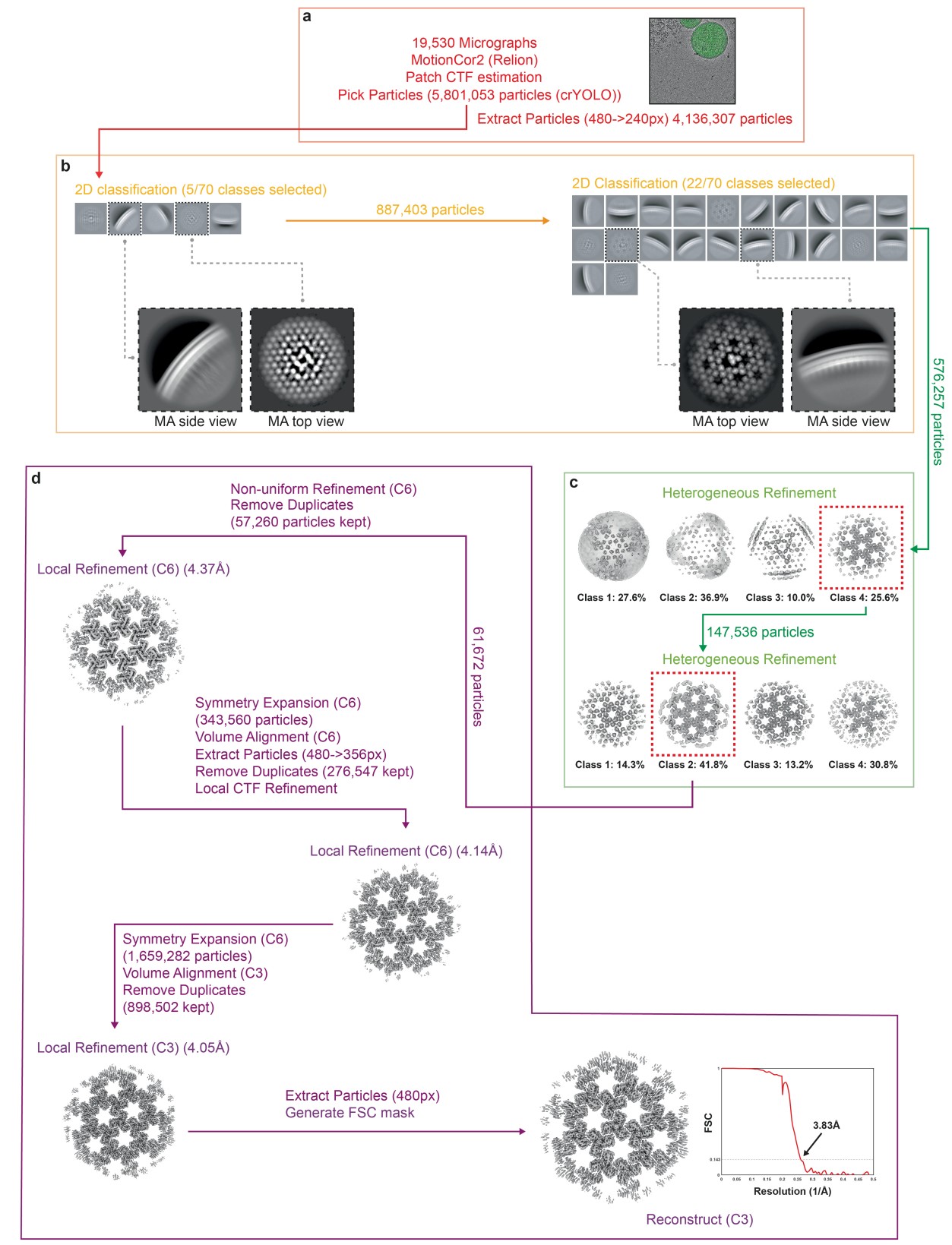

**Extended Data Fig. 6** | See next page for caption.

**Extended Data Fig. 6 | Single-particle cryo-EM processing for Mature (WT) MA lattice.** Mature (WT) virions are fully cleaved, and therefore contain a mature conical CA lattice and a mature MA layer[1]. **a**, EM micrograph pre-processing, particle picking and particle extraction. Micrographs were motion corrected using MotionCor2 in Relion and CTF estimation was performed using patch CTF estimation in cryoSPARC. Picking was performed using crYOLO, where a model was trained to pick arbitrary positions across the entire virus surface. **b**, Two rounds of 2D classification were employed to identify high quality side and top views of the MA layer (classes that showed top and side views of the mature CA were excluded). **c**, Two consecutive rounds of heterogenous refinement in 3D were used to further select for high quality particles containing the MA lattice. **d**, High quality particles, as determined by 2D classification and heterogenous refinement, were subject to global and then local 3D refinement which converged to a 4.37 Å reconstruction. The fact that MA forms a 2D lattice was then exploited to add particles at the expected positions of symmetry related neighbours, thereby expanding the set of particles (lattice expansion). At this stage the particles are centered on the six-fold axis. For each particle, six particles were created and were assigned positions and angles corresponding to the neighbouring six-fold axes as derived from the reconstruction. Following the lattice expansion an additional round of local refinement was then performed to improve alignment accuracy. The lattice expansion step was repeated once more, and the reconstruction was re-centred onto the three-fold trimer axis. Any duplicate particles were removed. The final particle set was reconstructed to generate a final reconstruction of the MA lattice at a resolution 3.83 Å.

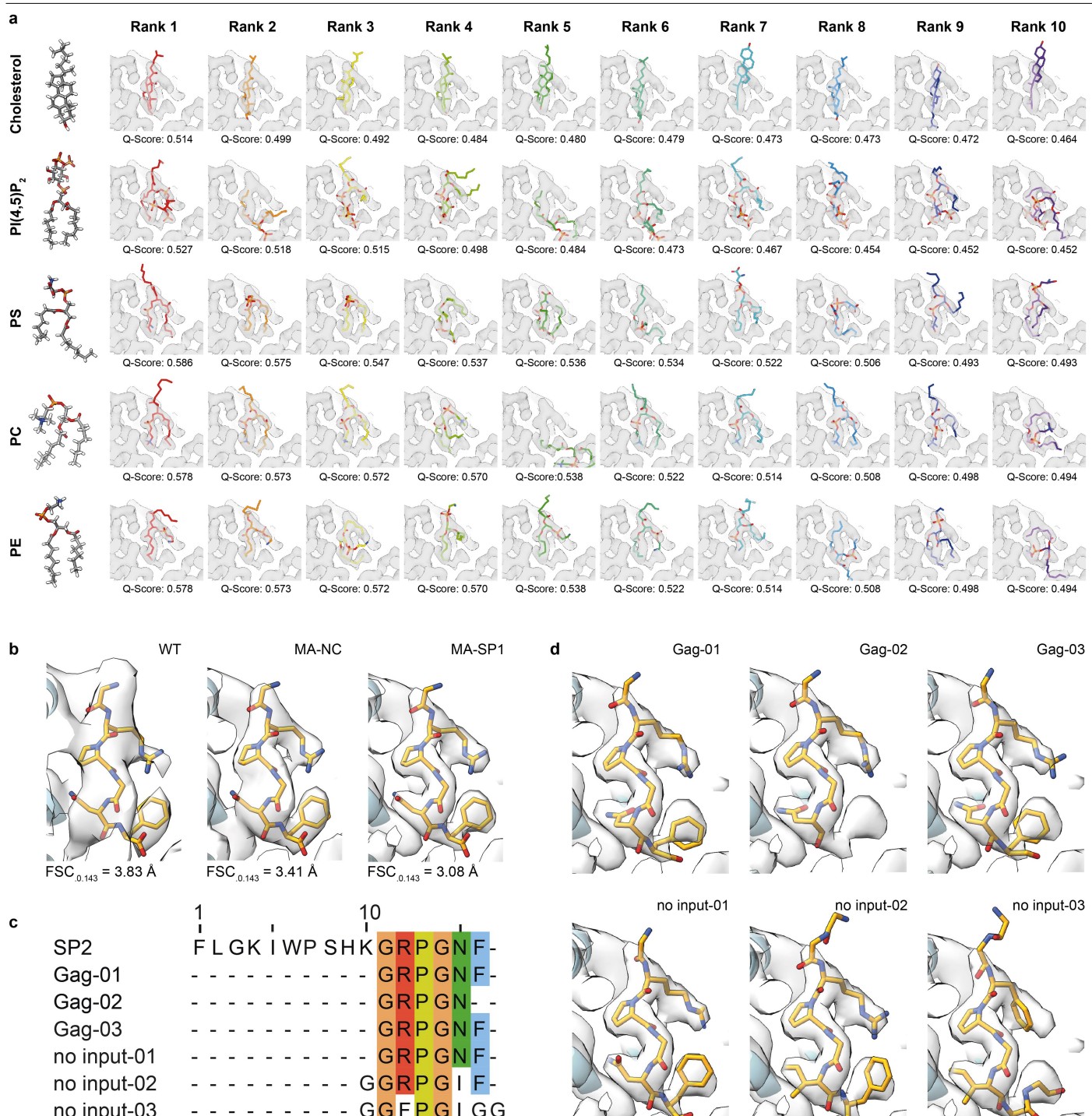

**Extended Data Fig. 7 | Automated fitting of lipid and protein into the mature MA side pocket ligand density. a**, Automated flexible ligand fitting was performed into the MA-SP1 ligand density using RosettaEmerald software[54]. The ligands tested were Cholesterol, PI(4,5)P$_2$, Phosphatidyl Serine (PS), Phosphatidyl Choline (PC) and Phosphatidyl Ethanolamine (PE). These ligands were chosen as they constitute the only lipids present in the HIV-1 viral membrane that are abundant enough to potentially completely occupy all MA binding sites (~2000), as according to Mücksch et al.[34]. The 10 resulting highest scoring fits (Q-Score) are shown in order of their rank. We identified no fit that could

adequately account for the observed side pocket density. **b**, Atomic models of SP2 C-terminal residues fit into the side pocket densities of all determined mature MA lattice reconstructions (WT, MA-NC and MA-SP1). **c**, Multiple sequence alignment of automatically predicted sequences in the side pocket MA density by ModelAngelo[60]. The first sequence is that of a full-length SP2 and the sequences below are either those predicted by ModelAngelo with the HIV-1 Gag sequence as input or with no sequence input. **d**, Atomic models built by ModelAngelo into the side pocket densities of the central MA trimer corresponding to the sequence predictions in c.

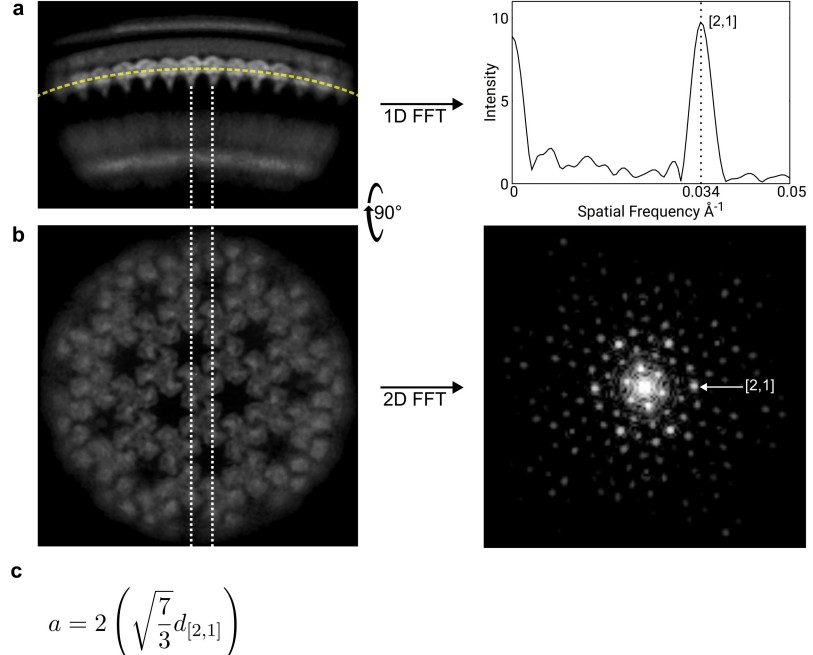

**a**

**b**

90°

1D FFT

2D FFT

[2,1]

[2,1]

**c**

$$a = 2\left(\sqrt{\frac{7}{3}}d_{[2,1]}\right)$$

**Extended Data Fig. 8 | Further details of 2D class Fourier analysis of HIV-1 cleavage mutants. a**, Side-view simulated projection of the mature MA map of the HIV-1 MA-SP1 cleavage mutant. The orientation of the map is close to that observed in the 2D class averages used in the 1D Fourier analysis of MA lattice. 1D fast Fourier transform (FFT) of pixel values extracted along the MA layer produces a single peak at a spatial frequency of 0.034 Å⁻¹ which corresponds to a mature lattice. **b**, Top-view (90-degree rotation of a. along the x-axis) simulated projection of the map. The MA molecules oriented along the dotted lines produce strong reflection at h,k [2,1] as visible in the 2D FFT of the simulated projection. This is the same [2,1] reflection observed in the 1D FFT of the MA side-view in a. **c**, Equation defining the relationship between the unit cell parameter *a* and the interplanar distance *d* of reflection h,k = [2,1].

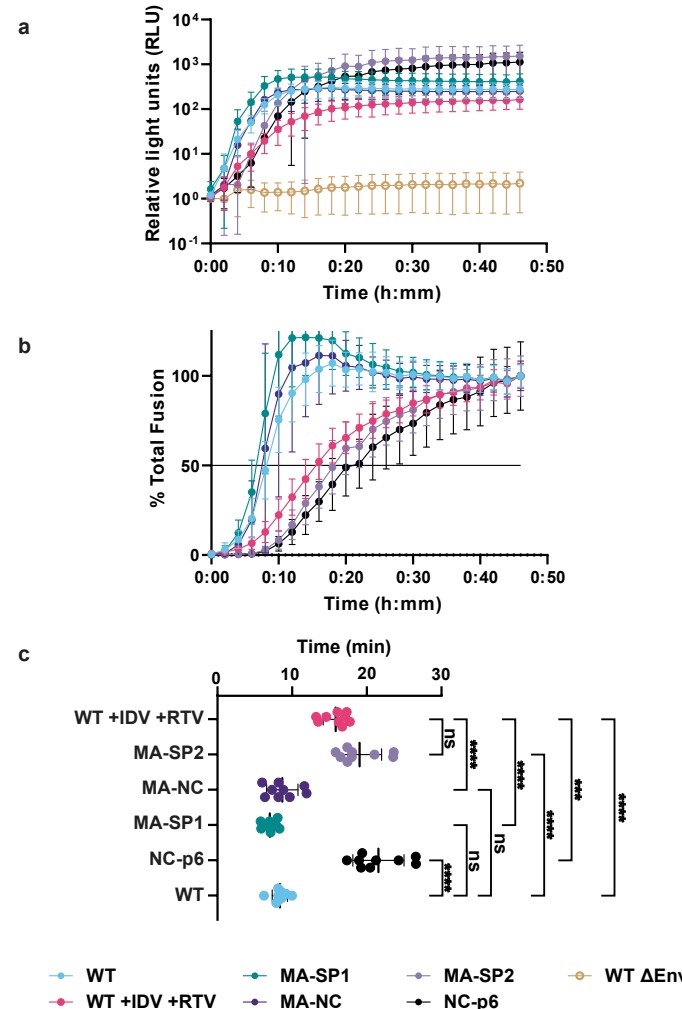

**a**, **b** (graphs)

**c** (histogram)

Legend:
- WT
- WT +IDV +RTV
- MA-SP1
- MA-NC
- MA-SP2
- NC-p6
- WT ΔEnv

**Extended Data Fig. 9 | SP2 release is required for WT-like fusion kinetics.**
**a**, Graph of total luciferase signal measured in relative light units for indicated
virus-like-particles (VLPs) all with WT JRFL Env. Blue = WT Gag VLPs; red = WT
Gag VLPs produced in the presence of the protease inhibitors Indinavir (IDV)
and Ritonavir (RTV); green = MA-SP1 Gag VLPs; dark purple = MA-NC Gag VLPs;
lilac = MA-SP2 Gag VLPs; black = NC-p6 Gag VLPs; yellow open circle = WT Gag
ΔEnv VLPs. Readings were measured every 2 min from 0 min to 46 min. **b**, Graph
of percent of total fusion for each sample in (**a**). Each sample is normalized with
its own luciferase signal at 46 min as 100%. Samples are labelled as in (**a**). Black
line represents 50% of total fusion. Plotted points in (**a**) and (**b**) are the means
from three biological replicates with three technical replicates per biological
replicate for an n = 9. Each biological replicate represents one aliquot from a
bulk preparation of viral particles. Error bars = mean ± s.d. **c**, Histogram of
fusion $T_{1/2}$ for each technical replicate in (**b**). Statistics were performed using an
ordinary one-way ANOVA test with Tukey's multiple comparisons tests. Brown-
Forsythe test and Bartlett's test were performed as corrections. Bars are
mean ± s.d. Specific mean ± s.d. values: 15.83 ± 1.671 (WT +IDV +RTV), 19.04 ± 2.943
(MA-SP2), 8.677 ± 2.101 (MA-NC), 7.024 ± 0.9382 (MA-SP1), 21.56 ± 3.418
(NC-p6), 8.357 ± 1.022 (WT). ns, $p \geq 0.10$; *** $p < 0.001$; **** $p < 0.0001$. Specific
$p$-values: WT +IDV +RTV vs NC-p6, $p = 0.0001$; WT +IDV +RTV vs MA-SP2,
$p = 0.1096$; WT vs MA-SP1, $p = 0.9386$; WT vs MA-NC, $p > 0.9999$. Panel (**c**) is
reproduced from Fig. 2d, but coloured in accordance with (**a**) and (**b**).

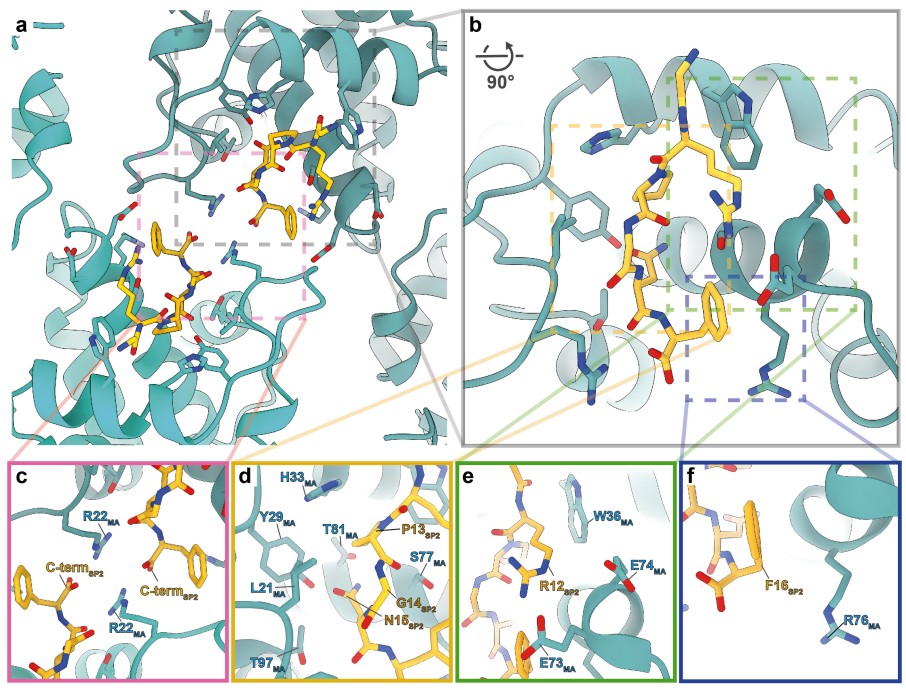

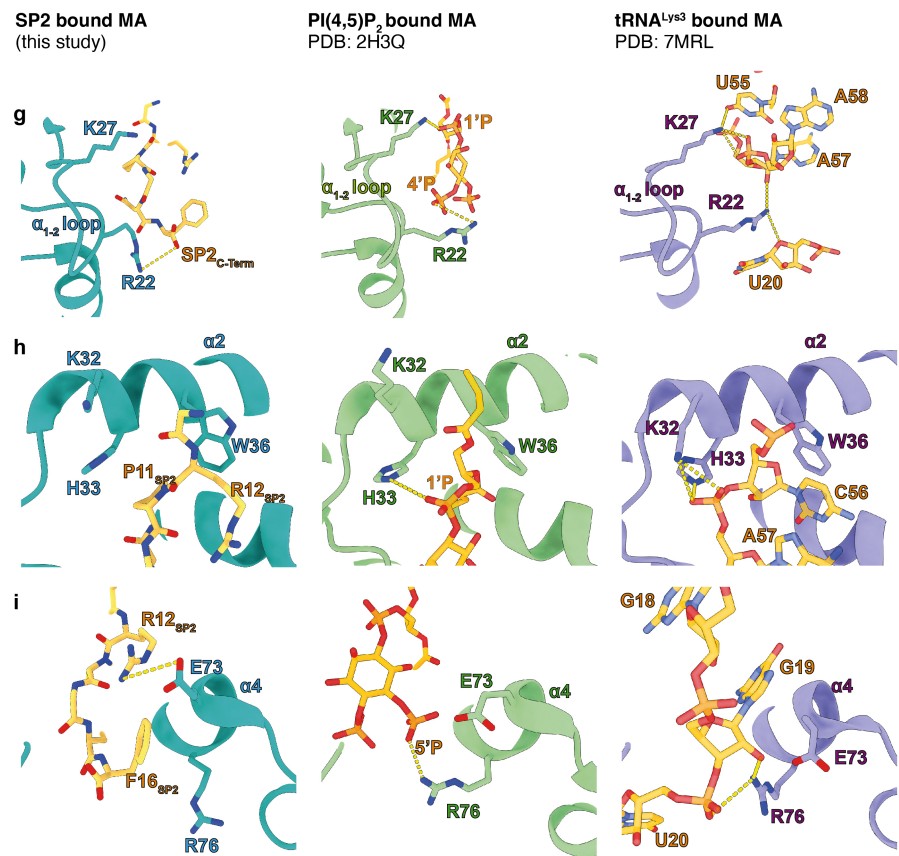

**Extended Data Fig. 10** | See next page for caption.

**Extended Data Fig. 10 | Structural details of the interaction between MA and SP2 and comparison with other side pocket ligands. a**, Atomic model of the dimeric interface of the mature MA lattice, with bound SP2, viewed from outside the virus (MA; Blue, SP2; orange). **b**, Rotated and zoomed in view of SP2 within the MA binding pocket from the side. **c**, Zoomed in view of (**a**). The C terminus of two adjacent SP2 chains and R22 from the two opposing MA molecules form a favourable electrostatic interaction network across the symmetry axis. This interaction likely results in the favouring of the mature MA lattice configuration upon SP2 binding to MA. **d-f**, Zoomed in views of b. **d**, Central SP2 binding motif residues P13, G14 and N15 sit in a hydrophobic cleft in the MA side pocket, formed by residues in helices 2, 4 and 5 as well as in the α-helix 1-2 loop. **e**, Binding of SP2 R12 is supported by a hydrophobic contact with W36 and an electrostatic interaction with residues E73 and E74. **f**, Binding of SP2 F16 is facilitated by a hydrophobic contact with R76. **g**, MA bound to SP2 (left. MA: pink, SP2: orange): R22 in the α-helix 1-2 loop forms an electrostatic interaction with the C-terminal carboxyl group of SP2, whereas K27 does not interact with SP2 (left). MA bound to PtdIns(4,5)P$_2$[12] (middle. MA green; PI(4,5)P$_2$ orange): R22 and K27 interact with the PtdIns(4,5)P$_2$ 4'-phosphate and bridging 1'-phosphodiester respectively. MA bound to tRNA (right. MA purple; tRNA$^{Lys3}$ orange): R22 forms an interaction with a 2'-OH ribose moiety (tA57) and a 4 O ribose position on the tRNA backbone (tU20) whereas K27 interacts with a backbone phosphate group (tA58) (right)[23]. **h**, Left: α-Helix 2 residues, H33 and W36, form π-stacking interactions with P11 and the R12 backbone amide of SP2 respectively. K32 does not interact with SP2. Middle: H33 forms an electrostatic interaction with the 1' phosphate of PI(4,5)P$_2$. Right: Both K32 and H33 form electrostatic interactions with backbone phosphate of the tRNA (tA57), whereas W36 forms a π stacking interaction with tC56. **i**, Left: α-Helix 4 residue E73 forms a salt bridge with R12 of SP2. F16 of SP2 packs against the acyl tail of R76. Middle: R76 forms an electrostatic interaction with the 5'-phosphate of PI(4,5)P$_2$. Right: R76 interacts with a backbone phosphate of tRNA as well as a ribose 2'-OH (tG16).

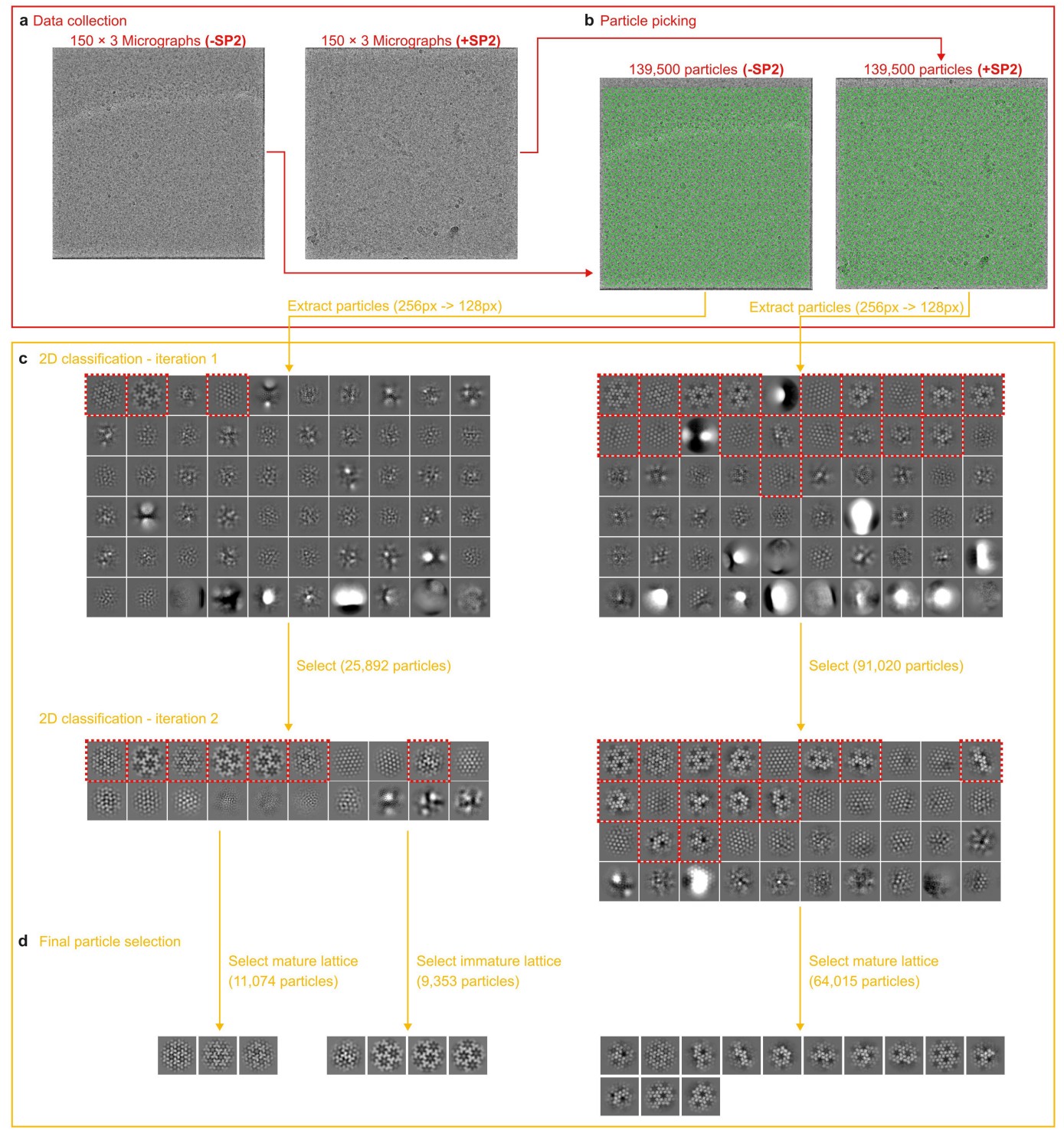

**Extended Data Fig. 11 | Data processing scheme of the 2D crystallography experiment. a**, 150 micrographs were collected randomly from each grid. Six grids were prepared simultaneously, three grids of the myrMA -SP2 control group and three grids of the myrMA +SP2 experimental group. **b**, Particles were picked randomly as a grid of points and extracted in cryoSPARC. **c**, The extracted particles were subjected to an initial round of 2D classification. 2D classes showing any kind of MA lattice were selected and subjected to a second round of 2D classification. **d**, Classes with mature and immature lattices were selected and particles belonging to these classes were used in the final statistical analysis.

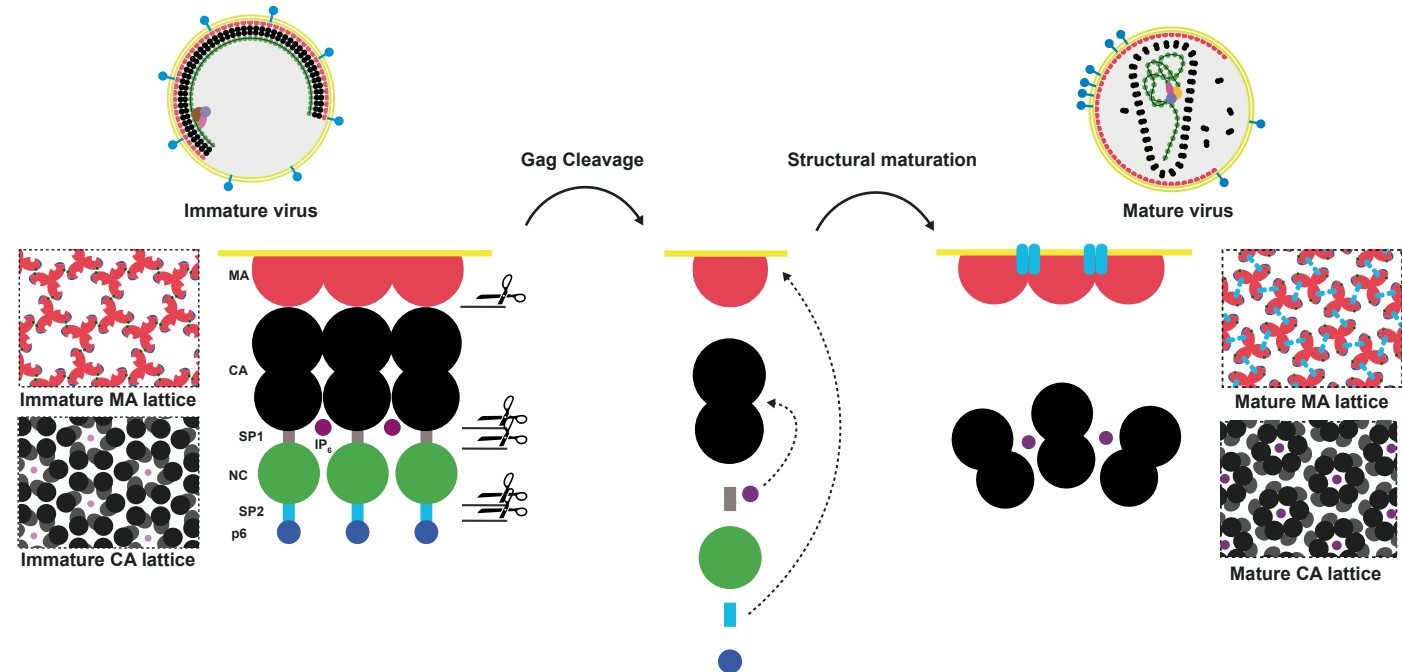

**Extended Data Fig. 12 | Schematic illustration of the mechanism of MA and CA maturation.** In the immature virus, MA and CA both form 'immature' hexameric lattices. Upon viral maturation, which is initiated by the activation of the viral protease (PR), Gag is cleaved into its constituent domains; MA, CA, SP1, NC, SP2 and p6. In the case of MA, subsequent structural transition to the mature lattice is induced upon binding of cleaved SP2 to MA. In the case of CA, cleavage of SP1 leads to the release of $IP_6$ which binds free CA to promote the formation of the mature CA lattice. Virus schematics adapted from Briggs and Kräusslich (2011)[70].

**Extended Data Table 1 | Summary table of Cryo-EM data collection and processing statistics**

|  | HIV-1<br>Immature (PR-)<br>EMD:52229 | HIV-1<br>MA-SP1<br>EMD:51769 | HIV-1<br>Mature (WT)<br>EMD:52221 | HIV-1<br>MA-NC<br>EMD:52222 |
|---|---|---|---|---|
| **Data collection and processing** |  |  |  |  |
| Magnification | 130,000x | 130,000x | 130,000x | 130,000x |
| Voltage (kV) | 300 | 300 | 300 | 300 |
| Electron exposure ($e^-/Å^2$) | 40 | 40 | 40 | 40 |
| Defocus range (μm) | -0.6 to -2.6 | -0.6 to -2.6 | -0.6 to -2.6 | -0.6 to -2.6 |
| Pixel size (Å) | 0.95 | 0.95 | 0.95 | 0.95 |
| Movies (no.) | 9,942 | 14,222 | 19,530 | 17,335 |
| Initial particle images (no.) | 3,568,755 | 5,704,512 | 5,801,053 | 7,238,540 |
| Symmetry imposed | *C3* | *C3* | *C3* | *C3* |
| Final particle images (no.)* | 174,245 | 5,486,693 | 889,805 | 7,745,404 |
| Map resolution (Å) | 5.83 | 3.08 | 3.83 | 3.41 |
| FSC threshold | 0.143 | 0.143 | 0.143 | 0.143 |

*Additional particles were derived during processing by lattice expansion (see Materials and Methods).

**Extended Data Table 2 | Summary table of model building and refinement statics of mature MA lattice in complex with SP2 (HIV-1 MA–SP1)**

|  | MA-SP1 Mature MA Lattice PDB:9H1P |
| --- | --- |
| **Refinement** |  |
| Initial model used | PDB: 2H3I |
| Model resolution (Å) | 3.24 |
| FSC threshold | 0.5 |
| Map sharpening $B$ factor (Å$^2$) | -167.7 |
| Model composition |  |
| Non-hydrogen atoms | 10668 |
| Protein residues | 1332 |
| Ligands | - |
| $B$ factors (Å$^2$) |  |
| Protein | 52.49 |
| Ligand | 0 |
| R.m.s. deviations |  |
| Bond lengths (Å) | 0.005 |
| Bond angles (°) | 1.025 |
|  |  |
| **Validation** |  |
| MolProbity score | 0.93 |
| Clashscore | 1.72 |
| Poor rotamers (%) | 0 |
| Ramachandran plot |  |
| Favored (%) | 100 |
| Allowed (%) | 0 |
| Disallowed (%) | 0 |
| Q-Score | 0.64 |

# Reporting Summary

## Statistics

For all statistical analyses, confirm that the following items are present in the figure legend, table legend, main text, or Methods section.

| n/a | Confirmed | |
|---|---|---|
| ☐ | ☒ | The exact sample size (*n*) for each experimental group/condition, given as a discrete number and unit of measurement |
| ☐ | ☒ | A statement on whether measurements were taken from distinct samples or whether the same sample was measured repeatedly |
| ☐ | ☒ | The statistical test(s) used AND whether they are one- or two-sided<br>*Only common tests should be described solely by name; describe more complex techniques in the Methods section.* |
| ☒ | ☐ | A description of all covariates tested |
| ☒ | ☐ | A description of any assumptions or corrections, such as tests of normality and adjustment for multiple comparisons |
| ☐ | ☒ | A full description of the statistical parameters including central tendency (e.g. means) or other basic estimates (e.g. regression coefficient) AND variation (e.g. standard deviation) or associated estimates of uncertainty (e.g. confidence intervals) |
| ☐ | ☒ | For null hypothesis testing, the test statistic (e.g. *F*, *t*, *r*) with confidence intervals, effect sizes, degrees of freedom and *P* value noted<br>*Give P values as exact values whenever suitable.* |
| ☒ | ☐ | For Bayesian analysis, information on the choice of priors and Markov chain Monte Carlo settings |
| ☒ | ☐ | For hierarchical and complex designs, identification of the appropriate level for tests and full reporting of outcomes |
| ☒ | ☐ | Estimates of effect sizes (e.g. Cohen's *d*, Pearson's *r*), indicating how they were calculated |

*Our web collection on statistics for biologists contains articles on many of the points above.*

## Software and code

Policy information about availability of computer code

| Data collection | Cryo-electron microscopy data were collected using the ThermoFisher EPU software versions 3.3 - 3.8. |
|---|---|
| Data analysis | All software used for data analysis is widely available.<br>Cryo-electron microscopy data were analyzed using the following software: crYOLO v1.7.6; Relion 4.0 and 5.0; cryoSPARC v3.3 and 4.4; CTFfind4; MotionCorr2.<br>Model building and refinement of the structural model was perfomed using UCSF Chimera 1.15; ChimeraX 1.3; ModelAngelo v1.0; Phenix v1.21 and ISOLDE 1.3.<br>Fourier analysis of the 2D class averages was performed in MATLAB v2022a (MathWorks).<br>Data were visualized using ChimeraX; Fiji/imageJ v1.64f; and MATLAB v2022a.<br>Automated lipid fitting was performed by RosettaEmerald (in Rosetta 2023.45).<br>Fusion data was processed using Mathematica and Excel. |

For manuscripts utilizing custom algorithms or software that are central to the research but not yet described in published literature, software must be made available to editors and reviewers. We strongly encourage code deposition in a community repository (e.g. GitHub). See the Nature Portfolio guidelines for submitting code & software for further information.

## Data

Policy information about availability of data

All manuscripts must include a data availability statement. This statement should provide the following information, where applicable:
- Accession codes, unique identifiers, or web links for publicly available datasets
- A description of any restrictions on data availability
- For clinical datasets or third party data, please ensure that the statement adheres to our policy

Structures determined by electron microscopy are deposited in the Electron Microscopy Data Bank under accession codes EMD-52229, EMD-51769, EMD-52221 and EMD-52222. The molecular model of the MA-SP1 MA lattice is deposited in the Protein Data Bank under accession code PDB:9H1P, for which PDB:2H3I was used as a starting model.

## Research involving human participants, their data, or biological material

Policy information about studies with human participants or human data. See also policy information about sex, gender (identity/presentation), and sexual orientation and race, ethnicity and racism.

| | |
|---|---|
| Reporting on sex and gender | NA |
| Reporting on race, ethnicity, or other socially relevant groupings | NA |
| Population characteristics | NA |
| Recruitment | NA |
| Ethics oversight | NA |

Note that full information on the approval of the study protocol must also be provided in the manuscript.

# Field-specific reporting

Please select the one below that is the best fit for your research. If you are not sure, read the appropriate sections before making your selection.

☒ Life sciences ☐ Behavioural & social sciences ☐ Ecological, evolutionary & environmental sciences

For a reference copy of the document with all sections, see nature.com/documents/nr-reporting-summary-flat.pdf

# Life sciences study design

All studies must disclose on these points even when the disclosure is negative.

| | |
|---|---|
| Sample size | Cryo-EM of HIV-1 cleavage mutants particles: at least two grids were prepared independently for each mutant. Thousands of HIV-1 particles were imaged and used for averaging and reconstruction. For further details see methods. Sample size was determined by the availability of microscope time for imaging, and because the sample size was sufficient to determine the structures and lattice parameenters.<br>2D crystallography: the number of grids imaged, number of grid squares imaged per grid and images per grid square were chosen to representatively sample the whole cryo-EM grid while having sufficient data to obtain good quality 2D classes.<br>Virus fusion experiment: Sample size was standard three biological replicates with three technical replicate per biological replicate. Each biological replicate represents one aliquot from a bulk preparation of viral particles. |
| Data exclusions | 2D crystallography and cryo-EM of cleavage mutants: Images were subjected to image classification procedures resulting in exclusion of some images. Image classification is a standard part of cryo-electron microscopy pipelines and are described in detail in the methods and Extended Data figures.<br>Virus fusion experiment: Data were not exluded. |
| Replication | cryo-EM data: At least two grids were prepared and imaged from each HIV-1 cleavage mutant sample. HIV-1 cleavage mutants samples were independently prepared and imaged more than once except for the MA-NC mutant which was prepared and imaged once. All replicates gave consistent results. For each sample, thousands of viruses were imaged.<br>2D crystallography: Grids were prepared in triplicate, and the results of all three replicates are included in the presented data.<br>Virus fusion experiment: Plotted points are from three biological replicates with three technical replicate per biological replicate. Each biological replicate represents one aliquot from a bulk preparation of viral particles. |
| Randomization | cryo-EM data: all datasets were randomly split to two independent halves and processed separately according to the "gold standard" procedure.<br>2D crystallography: Imaging locations within one grid square were chosen randomly. Subsequent analysis does not involve allocation to experimental groups<br>Virus fusion experiment: NA (experiment does not involve allocation to experimental groups) |

| Blinding | No blinding was performed. |
|---|---|

# Reporting for specific materials, systems and methods

We require information from authors about some types of materials, experimental systems and methods used in many studies. Here, indicate whether each material, system or method listed is relevant to your study. If you are not sure if a list item applies to your research, read the appropriate section before selecting a response.

## Materials & experimental systems

| n/a | Involved in the study |
|---|---|
| ☐ | ☒ Antibodies |
| ☐ | ☒ Eukaryotic cell lines |
| ☒ | ☐ Palaeontology and archaeology |
| ☒ | ☐ Animals and other organisms |
| ☒ | ☐ Clinical data |
| ☒ | ☐ Dual use research of concern |
| ☒ | ☐ Plants |

## Methods

| n/a | Involved in the study |
|---|---|
| ☒ | ☐ ChIP-seq |
| ☒ | ☐ Flow cytometry |
| ☒ | ☐ MRI-based neuroimaging |

## Antibodies

| Antibodies used | sheep αCA, polyclonal, 1:5,000 [in-house];<br>rabbit αNC, polyclonal, 1:400 [in-house];<br>Donkey anti-sheep IgG (H&L) Antibody DyLight™ 800 Conjugated, 1:10,000 [Rockland Cat# 613-731-168, RRID:AB_220181];<br>and Donkey anti-rabbit IgG IRDye 680 Conjugated 1:10,000 [LI-COR Biosciences Cat# 926-32223, RRID:AB_621845]. |
|---|---|
| Validation | Antibodies were used for Western blot only.<br>sheep αCA, polyclonal. The antiserum has been raised against purified recombinant HIV-1 CA. It was used for immunoblotting in previous studies (e.g., DOI: 10.1074/jbc.M109.027144 ; DOI: 10.1128/JVI.01704-12; DOI: 10.1128/JVI.00750-15), yielding the expected band patterns.<br>Rabbit αNC, polyclonal. The antiserum has been raised against purified recombinant HIV-1 NC. It was used for immunoblotting in previous studies (e.g., DOI: 10.1128/JVI.01704-12), yielding the expected band patterns. |

## Eukaryotic cell lines

Policy information about cell lines and Sex and Gender in Research

| Cell line source(s) | HEK293T (RRID:CVCL_0063), HEK293 ATCC number CRL-1573. |
|---|---|
| Authentication | HEK293T Genetic characteristics were confirmed by PCR-single-locus-technology (Eurofins; Promega power Plex 21 PCR Kit). HEK293 cells were authenticated by STR profiling. |
| Mycoplasma contamination | HEK293T and HEK293 cells were regularly tested for mycoplasma contamination and tested negative. |
| Commonly misidentified lines<br>(See ICLAC register) | No commonly misidentified cell lines were used in this study. |

## Plants

| Seed stocks | *Report on the source of all seed stocks or other plant material used. If applicable, state the seed stock centre and catalogue number. If plant specimens were collected from the field, describe the collection location, date and sampling procedures.* |
|---|---|
| Novel plant genotypes | *Describe the methods by which all novel plant genotypes were produced. This includes those generated by transgenic approaches, gene editing, chemical/radiation-based mutagenesis and hybridization. For transgenic lines, describe the transformation method, the number of independent lines analyzed and the generation upon which experiments were performed. For gene-edited lines, describe the editor used, the endogenous sequence targeted for editing, the targeting guide RNA sequence (if applicable) and how the editor was applied.* |
| Authentication | *Describe any authentication procedures for each seed stock used or novel genotype generated. Describe any experiments used to assess the effect of a mutation and, where applicable, how potential secondary effects (e.g. second site T-DNA insertions, mosiacism, off-target gene editing) were examined.* |

