## [Peer Review File · Nature]

The conserved HIV-1 spacer peptide 2 triggers matrix lattice maturation

Corresponding Author: Dr John Briggs

Version 0:

Reviewer comments:

Referee #1

(Remarks to the Author)

Stacey et al add a further twist to the complexity of HIV-1 assembly by showing that the spacer 2 peptide, after liberation by protease processing of Gag, binds in a groove on the side of each MA subunit and contributes interactions that stabilize the mature MA lattice. The work disproves one aspect of a previous interpretation of cryo-ET analysis of immature and mature particles -- viz, the suggestion that the density now assigned convincingly to spacer 2 was instead a fatty acyl chain from PIP2 in the membrane.

There are four complementary components to the effort: (1) production of a set of HIV-1 particles with various cleavage sites for the viral protease ablated; (2) single-particle cryo-EM analysis of the matrix layer in each case; (3) one validation of the interpretation by showing that spacer 2 induces formation of mature-like MA lattices on lipid monolayers; (4) a second validation of the interpretation by showing that fusion kinetics correspond to the virion-like rate only for the mutant particles in which spacer 2 is liberated from both NC and p6. The authors have achieved higher resolution than their previous, cryo-ET analysis, by using single-particle methods rather than subtomogram averaging.

The results convincingly support the revised interpretation, and my comments below are requests for some clarifications, as the MS is written only for HIV-1 experts, with the cryo-ET analysis written to impress and draw a cryo-EM reader into trying hard to understand it but giving up after a couple of hours because Suppl. Figs. 3-5 are incomprehensible (or rather poorly described -- each caption should be a page or more in length, and the reference back to Methods just isn't helpful).

1. I think it is fair to ask the authors to make it clear in the introduction that it is their own (or largely their own) model that they are revising. Until I checked the authorship of reference 22, I assumed that it was from another group. All they are doing is correcting their own previous interpretation, and the authors (or at least all who are on the previous paper) can be pleased that they have found a better interpretation of their earlier data. If they simply say that an earlier model, based both on some NMR results and on their cryo-ET data, suggested abstraction of a fatty acyl chain from the membrane, but their new data shows that the density is really spacer 2, then they could eliminate most of lines 76-94. I think they are overly hung up on disproving the lipid model, which seems odd in any case. In the same vein, I'd urge that in Suppl Fig. 1, they add a semi-transparent X over the now disproved step.

2. The particle nomenclature is a kind of lab in-joke, confusing except for a complete HIV-1 maven (I'm not). When I wanted to follow Suppl. Figs. 3-5, I got quite lost. All three have an immature CA layer, as far as I can tell, but which one(s) have immature or mature MA layers? Of course, I indeed checked by going back to the definitions, but by then I was quietly cursing the authors. If you're not an HIV-1 maven, then you also need help in understanding when CA remains immature (and not conical), even if MA matures because spacer 2 is liberated. No need to change the nomenclature -- I don't want to spoil the in-joke -- but in the captions of those figures (and any others that are relevant) just remind the reader: expect MA to be (mature/immature) and expect CA to be (mature cone or immature "sphere").

3. The single-particle treatment of the MA layer is a real tour de force of cryo-EM image analysis, but it is almost impossible to figure out from the description in Methods and from Suppl. Figs. 3-6 exactly how they proceeded. The authors do

themselves a disservice by minimizing what is a very insightful and challenging approach. (Indeed, a few more main-text sentences summarizing the analysis are in order -- simply saying "we did it in situ" undersells the analysis.) One almost has the impression that they wanted to make it seem so elaborate that no one else could dare think it through, let alone repeat it. Two specific points:

- (i) The description of particle picking is confusing. As I understood it, they trained crYOLO to pick "particles" of some box size all over the visible virions -- in other words, trained it to recognize "virus" from "ice" and peppered "virus" with many local "particles". But this description is my best guess from the the wording in Methods and may be wrong. So please clarify.
- (ii) The 2-D classes in Suppl Figs. 3-5 seem to include both "side" views from the virion periphery and "en face" views. For CA, there is no indication of subtraction of any other density, so how did the en face views arise without confusion at least from CA on the other side of the particle? Subtraction is indicated only in Suppl Fig 3 -- was it not done for the analyses in 4 and 5? What have I missed?

Bottom line: Nature should publish as soon as the authors have made some aspects of the MS clearer for a reader who is not an HIV-1 expert and who would like to follow how they teased out the MA lattice from the superposition with everything else in the virion. Should all be easy and quick.

Referee #2

(Remarks to the Author)

The manuscript by John Briggs and colleagues describes a functional role for the SP2 peptide that is generated during proteolytic maturation of HIV-1 Gag. They have previously shown that MA forms a hexameric protein lattice in immature virus particles that transitions into a mature MA lattice upon Gag cleavage. The work reported assigns a structural and functional role to SP2 generated by Gag cleavage.

The major findings are:

They now obtained a higher resolution map of the mature MA lattice, by applying in situ single-particle (SPA) analysis, which identified important additional density that no longer can be attributed to potential PIP2 binding. Instead they show that this extra density is compatible with binding the C-terminal six aa of SP2, thereby stabilizing the mature MA lattice. They constructed a series of Gag cleavage mutants, which confirmed that release of SP2 is important for MA maturation by comparing structures from immature MA with mature particles and MA-SP1 virus particles that also adopt a mature lattice. The importance of SP2 release was further confirmed by the analysis of cleavage site mutants MA-SP2, MA-NC, MA-SP1:NC-p6 and NC-p6. 2-D class averages of repetitive MA densities combined with measurements of the spacing of the MA layer by Fourier analysis confirmed that SP2 release is required for mature MA lattice formation.

They next demonstrated that the SP2 peptide can trigger the formation of mature MA lattices in vitro on planar lipid monolayers.

Finally, they present fusion assays that demonstrates that the release of SP2, the formation of mature MA lattice, accelerates membrane fusion kinetics.

In summary, the work is presented in a clear fashion and of high technical quality. The work is novel and of high interest to the structural virology community. Importantly, it closes a chapter on the function of the second peptide, P2, generated upon Gag proteolytic cleavage.

Although the work is quite complete and shows convincing evidence for the structural and functional role of SP2, the fate of its N-terminal 10 aa is less clear. The authors could show that SP2 full-length or the 10 N-terminal residues bind to membranes. Furthermore, it could be interesting to see whether the C-terminal 6 residues are sufficient for mature MA lattice formation in vitro.

Referee #3

(Remarks to the Author)

The authors here describe the surprising finding that the spacer peptide 2 (SP2) of HIV Gag binds to the matrix protein to induce maturation of the MA lattice. SP2 has not previously had a known function, and MA has been shown to bind to PI(4,5)P2 on the inner face of the plasma membrane. Results here show that cleavage of Gag releasing SP2 is critical for formation of the mature MA lattice, and that this is independent of the formation of the mature CA lattice. The mature MA lattice is shown to contribute to the speed of membrane fusion. Importantly, addition of SP2 in vitro led to formation of mature MA lattices on lipid monolayers, and recapitulated the density in the lattice that was seen in single particle cryoEM. The report is well-presented, uses outstanding techniques to derive its major findings, and is very convincing. This is a very meaningful result, as it illuminates an unsuspected contributor to maturation of Gag following cleavage, and reveals an important role for a cleavage peptide that was previously of unknown significance.

Use of statistics is appropriate. Conclusions fit the data shown and are robust. Writing is clear. References are appropriate.

Criticism (minor): please add an explanatory sentence in the text of the section beginning on line 174 providing the basics of the fusion assay referenced as reference 69 Yamamoto et al. A brief explanation is needed in order to facilitate interpretation of the kinetics of fusion shown. Presumably this was a VLP-cell fusion assay using the split nLuc system as described in one section of that paper.

Response to referees' comments:

Editorial remarks

We have made the following changes in response to editorial guidelines:

- Added accession numbers for all database depositions
- Uploaded the uncropped source data for Western blots as Supplementary Figure 1 and referred to this in the extended figure legend.
- Made minor changes to the introduction to shorten by approximately 200 words
- Made minor changes to the discussion to shorten by approximately 50 words
- Shortened all subheadings to 40 characters
- Shortened the legend of Figure 1 to 300 words
- Modified end note statements as per journal requirements
- Changed some figure annotations to lower case.

Please note that we have 14 extended data figures. Five of these figures are the processing schematic figures for the cryo-EM analyses. We do not see an easy way to get to 10 extended data figures while maintaining all the necessary EM methodology and ask for guidance on this point.

Referee #1 (Remarks to the Author):

Stacey et al add a further twist to the complexity of HIV-1 assembly by showing that the spacer 2 peptide, after liberation by protease processing of Gag, binds in a groove on the side of each MA subunit and contributes interactions that stabilize the mature MA lattice. The work disproves one aspect of a previous interpretation of cryo-ET analysis of immature and mature particles -- viz, the suggestion that the density now assigned convincingly to spacer 2 was instead a fatty acyl chain from PIP2 in the membrane.

There are four complementary components to the effort: (1) production of a set of HIV-1 particles with various cleavage sites for the viral protease ablated; (2) single-particle cryo-EM analysis of the matrix layer in each case; (3) one validation of the interpretation by showing that spacer 2 induces formation of mature-like MA lattices on lipid monolayers; (4) a second validation of the interpretation by showing that fusion kinetics correspond to the virion-like rate only for the mutant particles in which spacer 2 is liberated from both NC and p6. The authors have achieved higher resolution than their previous, cryo-ET analysis, by using single-particle methods rather than subtomogram averaging.

The results convincingly support the revised interpretation, and my comments below are requests for some clarifications, as the MS is written only for HIV-1 experts, with the cryo-ET analysis written to impress and draw a cryo-EM reader into trying hard to understand it but giving up after a couple of hours because Suppl. Figs. 3-5 are incomprehensible (or rather

poorly described -- each caption should be a page or more in length, and the reference back to Methods just isn't helpful).

We thank the reviewer for their helpful assessment of the manuscript. We have added much longer captions to Extended data figures 3-6 to better help the reader understand the image processing and made other changes to make this easier to follow, please see comments below.

1. I think it is fair to ask the authors to make it clear in the introduction that it is their own (or largely their own) model that they are revising. Until I checked the authorship of reference 22, I assumed that it was from another group. All they are doing is correcting their own previous interpretation, and the authors (or at least all who are on the previous paper) can be pleased that they have found a better interpretation of their earlier data. If they simply say that an earlier model, based both on some NMR results and on their cryo-ET data, suggested abstraction of a fatty acyl chain from the membrane, but their new data shows that the density is really spacer 2, then they could eliminate most of lines 76-94. I think they are overly hung up on disproving the lipid model, which seems odd in any case. In the same vein, I'd urge that in Suppl Fig. 1, they add a semi-transparent X over the now disproved step.

As requested by the reviewer, we have laid claim to the previous model by adding words like "our" etc to the introduction, and we have shortened the lines referred to by the reviewer. We have added a semi transparent X in Suppl. Fig. 1 as suggested.

2. The particle nomenclature is a kind of lab in-joke, confusing except for a complete HIV-1 maven (I'm not). When I wanted to follow Suppl. Figs. 3-5, I got quite lost. All three have an immature CA layer, as far as I can tell, but which one(s) have immature or mature MA layers? Of course, I indeed checked by going back to the definitions, but by then I was quietly cursing the authors. If you're not an HIV-1 maven, then you also need help in understanding when CA remains immature (and not conical), even if MA matures because spacer 2 is liberated. No need to change the nomenclature -- I don't want to spoil the in-joke -- but in the captions of those figures (and any others that are relevant) just remind the reader: expect MA to be (mature/immature) and expect CA to be (mature cone or immature "sphere").

We understand the reviewers frustration with the nomenclature, but we do not have a better one, and this has been used in previous literature. To help the reader we have added guidance on maturation state to the legend of Fig. 1, and we have added a key to the left side of Fig. 2 to indicate which mutants have immature and mature CA layers, while the right side indicates which mutants have immature and mature MA layers. We have included a sentence in the main text "*As expected, MA-SP2, MA-NC and MA-SP1 NC:p6 have immature CA lattices whereas NC-p6 has a mature CA lattice.*" As suggested, in the new captions of Extended data figures 3-6 we have clearly stated the maturation state of the CA layer and the MA layer.

3. The single-particle treatment of the MA layer is a real tour de force of cryo-EM image analysis, but it is almost impossible to figure out from the description in Methods and from

Suppl. Figs. 3-6 exactly how they proceeded. The authors do themselves a disservice by minimizing what is a very insightful and challenging approach. (Indeed, a few more main-text sentences summarizing the analysis are in order -- simply saying "we did it in situ" undersells the analysis.) One almost has the impression that they wanted to make it seem so elaborate that no one else could dare think it through, let alone repeat it. Two specific points:

As suggested, we added an additional main text sentence on the cryo-EM analysis.

(i) The description of particle picking is confusing. As I understood it, they trained crYOLO to pick "particles" of some box size all over the visible virions -- in other words, trained it to recognize "virus" from "ice" and peppered "virus" with many local "particles". But this description is my best guess from the the wording in Methods and may be wrong. So please clarify.

To clarify particle picking, we have reworded this section of the methods which now reads *"Picking models were trained to distinguish virus from background by manually and indiscriminately covering the complete surface of the virus with picks using the crYOLO boxmanager GUI (Extended Data Fig. 3-6)"*. We have added an inset illustrating picking positions to Extended data figures 3-6, and we have described particle picking in the new longer captions of the Extended data figures.

(ii) The 2-D classes in Suppl Figs. 3-5 seem to include both "side" views from the virion periphery and "en face" views. For CA, there is no indication of subtraction of any other density, so how did the en face views arise without confusion at least from CA on the other side of the particle? Subtraction is indicated only in Suppl Fig 3 -- was it not done for the analyses in 4 and 5? What have I missed?

For MA and CA, "en face" classes are found corresponding to either side of the particle (they have opposite hands) -- such classes were derived directly without subtraction. This is perhaps surprising, but the large datasets allow this. Subtraction was only performed for the immature PR- particles in Extended data figure 3. It was not necessary for the particles in Extended data Figures 4 and 5 (which also have immature CA lattices), because the mature MA lattice was sufficiently well ordered to align without subtraction. We have made this difference clear in the Extended data figure legends.

We have also made small changes throughout the methods section of the manuscript to try and make this easier for a reader to follow.

Bottom line: Nature should publish as soon as the authors have made some aspects of the MS clearer for a reader who is not an HIV-1 expert and who would like to follow how they teased out the MA lattice from the superposition with everything else in the virion. Should all be easy and quick.

Referee #2 (Remarks to the Author):

The manuscript by John Briggs and colleagues describes a functional role for the SP2 peptide that is generated during proteolytic maturation of HIV-1 Gag. They have previously shown that MA forms a hexameric protein lattice in immature virus particles that transitions into a mature MA lattice upon Gag cleavage. The work reported assigns a structural and functional role to SP2 generated by Gag cleavage.

The major findings are:

They now obtained a higher resolution map of the mature MA lattice, by applying in situ single-particle (SPA) analysis, which identified important additional density that no longer can be attributed to potential PIP2 binding. Instead they show that this extra density is compatible with binding the C-terminal six aa of SP2, thereby stabilizing the mature MA lattice.

They constructed a series of Gag cleavage mutants, which confirmed that release of SP2 is important for MA maturation by comparing structures from immature MA with mature particles and MA-SP1 virus particles that also adopt a mature lattice.

The importance of SP2 release was further confirmed by the analysis of cleavage site mutants MA-SP2, MA-NC, MA-SP1:NC-p6 and NC-p6. 2-D class averages of repetitive MA densities combined with measurements of the spacing of the MA layer by Fourier analysis confirmed that SP2 release is required for mature MA lattice formation.

They next demonstrated that the SP2 peptide can trigger the formation of mature MA lattices in vitro on planar lipid monolayers.

Finally, they present fusion assays that demonstrates that the release of SP2, the formation of mature MA lattice, accelerates membrane fusion kinetics.

In summary, the work is presented in a clear fashion and of high technical quality. The work is novel and of high interest to the structural virology community. Importantly, it closes a chapter on the function of the second peptide, P2, generated upon Gag proteolytic cleavage.

Although the work is quite complete and shows convincing evidence for the structural and functional role of SP2, the fate of its N-terminal 10 aa is less clear. The authors could show that SP2 full-length or the 10 N-terminal residues bind to membranes. Furthermore, it could be interesting to see whether the C-terminal 6 residues are sufficient for mature MA lattice formation in vitro.

We thank the reviewer for their positive assessment of the manuscript. The reviewer has identified two interesting areas for future experiments relating to the contributions of the N- and the C-terminal parts of SP2 to its function. Characterising the membrane-binding properties of the peptide requires the application of different experimental methods and goes substantially beyond the scope of the current study. While we have some data, the

SENTENCE REDACTED

in vivo validation of such results is very difficult because of the difficulty in interpreting mutagenesis experiments for the SP2 region which encodes two reading frames, two cleavage sites and an ribosomal frameshifting signal. We believe that the proposed experiments better represent the beginning of a future study that unpicks the contributions to SP2 function of the N- and C-terminal parts of the peptide, and indeed of individual residues of the peptide.

Referee #3 (Remarks to the Author):

The authors here describe the surprising finding that the spacer peptide 2 (SP2) of HIV Gag binds to the matrix protein to induce maturation of the MA lattice. SP2 has not previously had a known function, and MA has been shown to bind to PI(4,5)P2 on the inner face of the plasma membrane. Results here show that cleavage of Gag releasing SP2 is critical for formation of the mature MA lattice, and that this is independent of the formation of the mature CA lattice. The mature MA lattice is shown to contribute to the speed of membrane fusion. Importantly, addition of SP2 in vitro led to formation of mature MA lattices on lipid monolayers, and recapitulated the density in the lattice that was seen in single particle cryoEM. The report is well-presented, uses outstanding techniques to derive its major findings, and is very convincing. This is a very meaningful result, as it illuminates an unsuspected contributor to maturation of Gag following cleavage, and reveals an important role for a cleavage peptide that was previously of unknown significance.

Use of statistics is appropriate. Conclusions fit the data shown and are robust. Writing is clear. References are appropriate.

Criticism (minor): please add an explanatory sentence in the text of the section beginning on line 174 providing the basics of the fusion assay referenced as reference 69 Yamamoto et al. A brief explanation is needed in order to facilitate interpretation of the kinetics of fusion shown. Presumably this was a VLP-cell fusion assay using the split nLuc system as described in one section of that paper.

We thank the reviewer for their positive assessment of the manuscript. As requested, we have added an explanatory sentence into the text to provide the basics of the fusion assay. *“We used a split nanoluciferase complementation assay³⁰ with one enzyme fragment incorporated into virus-like particles and the other into target cells. Particles were added to the target cells, luminescence/fusion was measured over time, and the time at which each sample reaches 50% of total fusion was calculated ($T_{1/2}$). We found that membrane fusion of immature particles (produced in the presence of protease inhibitors) was two-fold slower than that observed in WT mature particles (Fig. 2d, Extended Data Fig. 9).”*

Other changes made to the manuscript

We have made the following additional changes to the manuscript:

Removed original reference 40 and corrected original supplementary reference 70.